# Cis-trans isomerization of peptoid residues in the collagen triple-helix

Rongmao Qiu[1,2], Xiaojing Li[1,2], Kui Huang[1,2], Weizhe Bai[3], Daoning Zhou[1,2], Gang Li[3,5] ✉, Zhao Qin [4,5] ✉ & Yang Li [1,2,3,5] ✉

Cis-peptide bonds are rare in proteins, and building blocks less favorable to the trans-conformer have been considered destabilizing. Although proline tolerates the cis-conformer modestly among all amino acids, for collagen, the most prevalent proline-abundant protein, all peptide bonds must be trans to form its hallmark triple-helix structure. Here, using host-guest collagen mimetic peptides (CMPs), we discover that surprisingly, even the cis-enforcing peptoid residues (N-substituted glycines) form stable triple-helices. Our interrogations establish that these peptoid residues entropically stabilize the triple-helix by pre-organizing individual peptides into a polyproline-II helix. Moreover, noting that the cis-demanding peptoid residues drastically reduce the folding rate, we design a CMP whose triple-helix formation can be controlled by peptoid cis-trans isomerization, enabling direct targeting of fibrotic remodeling in myocardial infarction in vivo. These findings elucidate the principles of peptoid cis-trans isomerization in protein folding and showcase the exploitation of cis-amide-favoring residues in building programmable and functional peptidomimetics.

The combination of three dihedral angles, φ, ψ, and ω, dictates the backbone folding of all proteins (Fig. 1a). The partial double bond character confines the planar peptide bond to either the trans or cis configuration (denoted by ω = 180° or 0°, respectively) in folded proteins. Although the isomerization of the cis- and trans-peptide bond can lead to crucial conformational changes in local and even global protein structure for life-dependent functions[1-3], the vast majority of all peptide bonds in natural proteins are trans, while cis-peptide bonds are rare, accounting for less than 0.1% (Fig. 1b)[4,5]. This phenomenon has provided the rationale behind the development of proteomimetics relying on trans-amide-favoring residues, while the structural effect of a cis-amide-favoring residue remains largely underexplored to date.

Compared to other natural amino acids that strongly prefer the trans conformation due to steric hindrance, proline is slightly more open to the cis-peptide bond because of its distinctive amine-bonded cyclic sidechain structure (Fig. 1b)[4], making a proline-rich sequence a suitable model to interrogate the significance of cis-amides in protein folding. Collagen, the most abundant protein in vertebrates and the main component of the extracellular matrix (ECM), is the most prevalent proline-rich protein in nature. Collagen has a hallmark repeating sequence of Gly-Xaa-Yaa triplets, where approximately 33% of the Xaa and Yaa locations are occupied by proline (Pro, P) or hydroxyproline (Hyp, O) in humans[6]. The resulting folding motif of collagen consists of three left-handed polyproline-II helix (PPII) chains intertwined into a unique right-handed triple-helix structure. Despite the high proline content, all peptide bonds in collagen must be trans for the triple-helix to form[7], implying that the cis-peptide bond of proline may be an obstacle to the collagen structure and folding[8,9].

[1]Guangdong Provincial Engineering Research Center of Molecular Imaging, the Fifth Affiliated Hospital, Sun Yat-sen University, Zhuhai, Guangdong 519000, China. [2]Guangdong-Hong Kong-Macao University Joint Laboratory of Interventional Medicine, the Fifth Affiliated Hospital, Sun Yat-sen University, Zhuhai, Guangdong 519000, China. [3]Cardiac Surgery and Structural Heart Disease Unit of Cardiovascular Center, the Fifth Affiliated Hospital, Sun Yat-sen University, Zhuhai, Guangdong 519000, China. [4]Department of Civil & Environmental Engineering, College of Engineering & Computer Science, Syracuse University, Syracuse, New York 13244, USA. [5]These authors jointly supervised this work: Gang Li, Zhao Qin, Yang Li. ✉e-mail: gangli73@163.com; zqin02@syr.edu; liyang266@mail.sysu.edu.cn

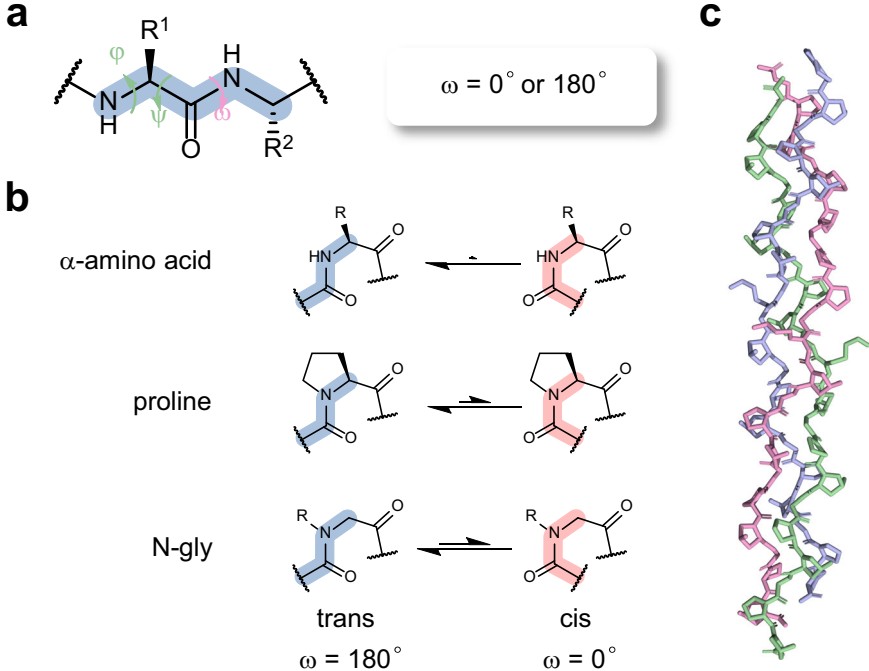

**Fig. 1 | The amide cis-trans isomerization and the collagen triple-helix. a** The three dihedral angles for protein backbone folding: φ, ψ, and ω. **b** The cis-trans isomerization of the peptide bond for amino acid, proline, and peptoid residues (or *N*-substituted glycine, N-gly). **c** The crystallography structure of a collagen mimetic peptide (CMP) triple-helix (PDB: 7jx4).

For decades, synthetic collagen mimetic peptides (CMPs), typically featuring the (GPO)$_n$ or (POG)$_n$ ($n$ = 6-10) sequences[10], have been widely used as models to elucidate the structural features that contribute to collagen stability (Fig. 1c)[11–18], including the effects of the Pro cis-trans isomerization[13,19,20]. By introducing proline derivatives with C$^\gamma$ electron-withdrawing groups [e.g., (4*S*)-fluoroproline (flp), (4*S*)-chloroproline (clp), (4*S*)-azidoproline (Azp)] to the Xaa or Yaa position of CMPs, pioneering scientists found that modified proline residues with a higher cis-amide propensity tend to destabilize the CMP triple-helix[12,18,21]. Nonetheless, such proline derivatization not only leads to an increased cis:trans ratio of the amide but also can affect the pyrrolidine ring pucker (i.e., C$^\gamma$-*endo* v.s. C$^\gamma$-*exo*) and the φ, ψ dihedral angles[13,18,22]. Therefore, the effects of the cis-trans preference on the collagen triple-helix structure remain to be explicitly defined, both thermodynamically and kinetically. *N*-substituted glycines (N-glys, or peptoid residues) are unnatural proline analogs with a much wider range of cis-amide propensity than modified prolines (Fig. 1b)[23]. We wonder how cis-trans isomerization of a peptoid residue lacking the φ, ψ constraints affects collagen folding. More interestingly, can a protein folding motif, such as the collagen triple-helix, where all peptide bonds must be trans, accommodate a cis-demanding residue?

Here we systematically establish the correlation between the cis-trans propensity of the Xaa-positioned peptoid residues and the collagen triple-helix structure. Surprisingly, we find that the cis-amide demanding peptoid residues may still form stable triple-helices, but drastically reduce the folding rate. Taking advantage of this discrepancy between the thermodynamic and kinetic properties, we design a cis-amide featuring CMP whose triple-helix folding can be controlled by the cis-trans isomerization of its peptoid residues; this CMP enables us to target the damaged collagen in the fibrotic lesion of myocardial infarction in vivo and perform 3D fluorescence imaging of fibrosis throughout the infarcted heart. Beyond collagen, our work contributes to the elucidation of the structural principles of the cis-amide-favoring residues in protein folding and demonstrates that peptoid residues with cis-amide propensities can be exploited for designing future, programmable proteomimetics with specific structures and activities.

## Results

### Effect of the cis-amide propensy of N-glys on CMP stability

To investigate the effect of cis-amide propensity on the stability of collagen triple-helix, we introduced a series of peptoid residues at the central X position of a CMP host-guest peptide with the sequence of Ac-(GlyProHyp)$_3$-Gly-X-Hyp-(GlyProHyp)$_3$-NH$_2$ (designated as X-CMP, Fig. 2a, b)[24]. We measured the circular dichroism (CD) spectra of the X-CMPs (Supplementary Fig. 1a, b) and assessed the X-CMPs' triple-helical stability (measured by their melting temperature $T_m$ values) via thermal unfolding experiments monitored by CD at 225 nm under a heating rate of 0.5 °C min$^{-1}$ (see Supplementary Methods, Fig. 2c, Supplementary Fig. 1c, d)[25]. To measure the inherent cis-trans propensity of each N-gly residue X in its monomer state, we synthesized each model compound Ac-X-OMe and calculated the ratio between its integrated cis- and trans-related peaks in the $^1$H NMR spectrum as its $K_{cis/trans}$ value (see Supplementary Methods, Fig. 2a, d, and Supplementary Information, Section 3). The Ac-X-OMe model has been extensively utilized for decades to study the cis-trans propensity of modified prolines and peptoid residues to correlate with the conformation of the sequences featuring these residues, including CMPs[12,21,23,26–30]. As shown in Fig. 2c, d, we found that the Ac-X-OMe compounds' $K_{cis/trans}$ values for the peptoid residues we previously reported (i.e., Sar, Nchx, Nleu, Nphe, Nasn, Nlys) ranged from 0.40 to 0.80 and were all higher than Pro ($K_{cis/trans}$ = 0.26), whereas the $T_m$ values of their X-CMPs were almost all greater than that of Pro-CMP, with Nchx-CMP being the most stable ($T_m$: 64 °C). These data implied that peptoid residues with cis-amide propensity greater than proline can also stabilize the helix.

Although the peptoid residues in Fig. 2d have $K_{cis/trans}$ values higher than Pro, after all, they still favor the trans-amide bond with their $K_{cis/trans}$ values all below 1. So we incorporated in the X-CMP host a group of N-gly guests whose cis-amide configuration is strongly promoted due to hydrogen bonding or n→π*$_{Ar}$ electronic interactions between the sidechains and the backbone carbonyl (Fig. 2e)[31,32]. Their $K_{cis/trans}$ values ranged from 3.78 to 10.85, which are about 15–42 times higher than that of Pro. However, we found that these cis-demanding

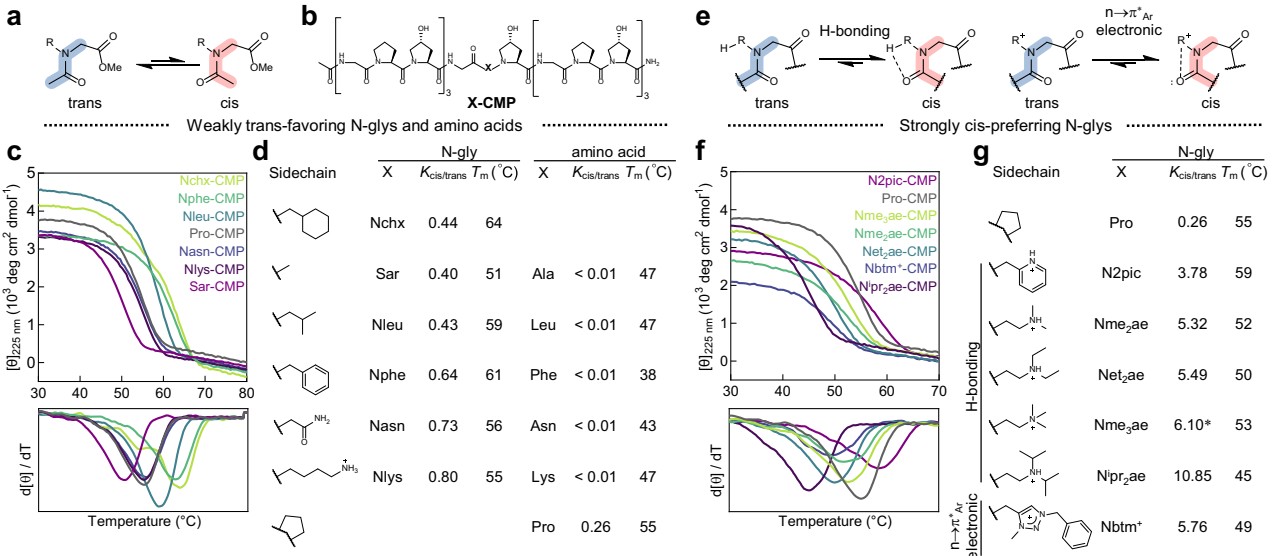

**Fig. 2 | The triple-helix stability of X-CMP featuring a weakly trans-amide favoring (Pro < $K_{cis/trans}$ < 1) or strongly cis-amide preferring ($K_{cis/trans}$ > 1) peptoid residue X. a** The cis-trans isomerization of the peptide bond for an N-gly model compound Ac-X-OMe, which was used to calculate the $K_{cis/trans}$ value for residue X. **b** Chemical structure of X-CMP, a host-guest collagen mimetic peptide with the sequence of (GlyProHyp)$_7$, where the central X-position Pro was substituted. **c** The CD thermal unfolding curves (top) and their first derivatives (bottom) of the X-CMPs featuring Pro or a weakly trans-favoring N-gly ($K_{cis/trans}$ < 1) as the guest X unit. As temperature increases, an X-CMP triple-helix will unfold into single chains, shown as a drop in CD signal, where the mid-point of this two-state transition (i.e., the lowest point of the derivative curve) is defined as the melting temperature ($T_m$, in °C), indicating the thermal stability of the X-CMP. **d** Triple-helical stabilities of X-CMPs featuring N-gly or amino acid residues with variable $K_{cis/trans}$ values. **e** Cis-amide induction caused by hydrogen bonding or $n \rightarrow \pi^*_{Ar}$ electronic interactions. **f** The CD thermal unfolding curves (top) and their first derivatives (bottom) of the X-CMPs featuring a series of strongly cis-amide favoring N-glys ($K_{cis/trans}$ > 1). **g** The structures and $K_{cis/trans}$ values for the strongly cis-amide favoring N-glys, as well as the $T_m$ values of their corresponding X-CMPs in PBS solutions (except for N2pic-CMP, which was measured in 1 mM HCl). *: value cited from ref. 31.

N-glys can still form stable X-CMP triple-helices ($T_m$: 45–59 °C, Fig. 2f, g). For instance, the $K_{cis/trans}$ value of N2pic was 3.78, which was almost 15 times the value of Pro, but the $T_m$ value of N2pic-CMP was 4 °C higher than Pro-CMP; with a $K_{cis/trans}$ value of 6.10 (almost 23 times the value of Pro), Nme$_3$ae made a CMP triple-helix just 2 °C less stable than Pro-CMP; even with the highest $K_{cis/trans}$ value of 10.85, N$^i$pr$_2$ae-CMP was only 10 °C less stable than Pro-CMP (Fig. 2g), but its stability was comparable to Lys-CMP ($T_m$: 47 °C) and Asn-CMP ($T_m$: 43 °C, Fig. 2d). Moreover, the destabilization of N$^i$pr$_2$ae and Net$_2$ae compared to Pro may be partially attributed to the steric hindrance from the bulky sidechains, a factor that further lessens the contribution from the cis-trans isomerization.

Next, we wondered how the stability of an X-CMP triple-helix changes if the cis-amide propensity of the X-peptoid residue is altered. Noting that the preferred cis-amide configuration of residues N2pic, Nme$_2$ae, Net$_2$ae, and N$^i$pr$_2$ae critically depends on their intramolecular hydrogen bonding between the protonated sidechains and the amide carbonyl[31,33], we first compared the $K_{cis/trans}$ values of their Ac-X-OMe compounds with and without protonation (Fig. 3). Following deprotonation, these strongly cis-demanding peptoid residues had become modestly trans-favoring, with their $K_{cis/trans}$ values drastically decreasing by 5.2 to 19 times (Fig. 3b). However, the $T_m$ values of the corresponding X-CMPs at the neutral state only increased by 0–5 °C compared to themselves at the protonated state (Fig. 3a, b). Indeed, the stability of N$^i$pr$_2$ae-CMP did not change at all even with a nearly 19-fold drop in $K_{cis/trans}$ value (10.85→0.58). These results showcased that converting the cis-trans preference of the peptoid residue X may not strongly affect the triple-helical stability of the X-CMP.

## PPII pre-organization of the peptoid residues

To gain insight into the foundation of the triple-helix stability of the N-glys, we measured a series of thermodynamic parameters for the folding of triple-helical X-CMPs by differential scanning calorimetry (DSC, see Supplementary Methods)[34,35]. We envisioned that if the N-gly can help pre-organize the backbone of each CMP single-strand into the desired conformation for the triple-helix formation, the entropy cost of the folding becomes more favorable (i.e., a lower $-T\Delta S$ value); otherwise, if the cis-trans preference of the X residue hinders the pre-organization of the CMP single-strand, the entropy cost of the folding should increase[35,36]. Our data showed that the entropy costs of all the N-gly X-CMPs we examined were lower than that of Pro-CMP, despite the stronger cis propensities of these peptoid residues compared to Pro (Fig. 4a and Supplementary Fig. 2). Also, the $\Delta S$ values of the peptoid residues did not show a correlation with their cis-trans propensity; for example, from Nchx-CMP to Nme$_2$ae-CMP, their $-T\Delta S$ values did not increase with the $K_{cis/trans}$ values of the N-glys (Fig. 4a). In addition, although the natural amino acid residues have a predominant preference for the trans-amide, the $-T\Delta S$ values of their corresponding CMPs (e.g., Leu-CMP and Lys-CMP) were still almost identical to that of Pro-CMP (Fig. 4a). Together, these thermodynamic data demonstrated that the N-gly residues we tested may reduce the entropy cost of the triple-helix folding compared to Pro regardless of their cis-trans propensity. Moreover, we compared the DSC data of N2pic- and Nme$_2$ae-CMP at different pH levels (Supplementary Fig. 3). It is interesting to note that when the two X-residues became over 5 times more cis-promoting as a result of sidechain protonation, their CMPs' $-T\Delta S$ values became even more favorable for the triple-helix formation, implying that the cis propensity is not the main entropic factor contributing to the folding.

Next, we performed a classical molecular dynamics (MD) simulation of the X-CMPs (see Supplementary Methods)[37–39]. In our simulated CMP triple-helices, both the weakly trans-favoring Nlys ($K_{cis/trans}$ = 0.80) and the strongly cis-biased Nme$_2$ae ($K_{cis/trans}$ = 5.3) have always maintained the trans configuration ($\omega \approx \pm 180$, Fig. 4b) with no backbone distortion. These results suggested that the cis-favoring N-glys could also adopt the trans configuration in a triple-helical protein folding.

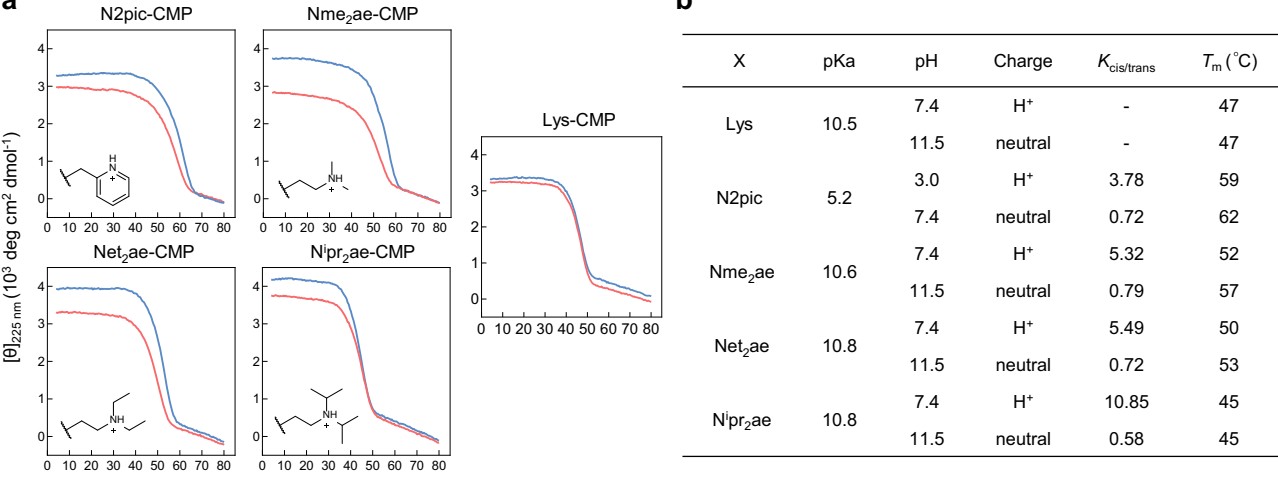

**b**

| X | pKa | pH | Charge | $K_{cis/trans}$ | $T_m$ (°C) |
|---|---|---|---|---|---|
| Lys | 10.5 | 7.4 | H$^+$ | - | 47 |
| | | 11.5 | neutral | - | 47 |
| N2pic | 5.2 | 3.0 | H$^+$ | 3.78 | 59 |
| | | 7.4 | neutral | 0.72 | 62 |
| Nme$_2$ae | 10.6 | 7.4 | H$^+$ | 5.32 | 52 |
| | | 11.5 | neutral | 0.79 | 57 |
| Net$_2$ae | 10.8 | 7.4 | H$^+$ | 5.49 | 50 |
| | | 11.5 | neutral | 0.72 | 53 |
| N$^i$pr$_2$ae | 10.8 | 7.4 | H$^+$ | 10.85 | 45 |
| | | 11.5 | neutral | 0.58 | 45 |

**Fig. 3 | Reversing the cis-trans propensity of the X peptoid residues barely changes their triple-helix stability. a** Thermal unfolding curves of N2pic-CMP, Nme$_2$ae-CMP, Net$_2$ae-CMP, and N$^i$pr$_2$ae-CMP in the protonated (red) and neutral (blue) states. **b** The sidechain pKa and the $K_{cis/trans}$ values of residue X, as well as the $T_m$ values of the corresponding X-CMP in the protonated or neutral state under different pH levels. While the $K_{cis/trans}$ values of N2pic, Nme$_2$ae, Net$_2$ae, and N$^i$pr$_2$ae drastically decreased by 5–19 folds after deprotonation, the triple-helical stabilities of their X-CMPs varied only slightly (0–5 °C).

Interestingly, throughout the relaxation process the deviations of the φ, ω dihedral angles adopted by Nlys and Nme$_2$ae within the simulated triple-helix were similar to and even slightly smaller than those of Pro, whereas their amino acid isomer Lys offers a wider φ angle distribution than Pro (Fig. 4b). More MD simulation results of X-CMPs also indicated the same trend (Supplementary Fig. 4a). These results seemed to indicate that the φ angles of Pro and various N-glys are restricted within a smaller range (i.e., less flexible backbone), while the φ angle deviation of the amino acids are larger and their CMPs have lower $T_m$ values (Supplementary Fig. 4b). These simulation results suggested that the folding of N-glys, even the strongly cis-biased Nme$_2$ae and Nme$_3$ae, can have low conformational perturbations inside the triple-helical peptide backbone and be more stable than many amino acids.

Triple-helix formation is difficult due to the need to assemble three chains in the absence of a hydrophobic core to stabilize the resulting association[13]. Therefore, the CMP triple-helix folding requires each peptide chain to adopt a stable PPII backbone. To verify whether the peptoid residues can promote the PPII conformation of a peptide, we adopted a series of host-guest peptides (X-PP5, sequence: Ac-Gly-ProPro-X-ProProGlyTyr-NH$_2$) and measured their CD spectra (see Supplementary Methods, Fig. 4c–e, and Supplementary Information, Section 4). This host-guest model has been previously exploited to systematically assess the PPII propensity of all natural amino acid residues[40,41]. The CD spectra of all our X-PP5 peptides exhibited a positive band of 220–230 nm and a negative band of 200–210 nm (Fig. 4d); this CD profile has been accepted by many as the hallmark of PPII conformation[42–45]. Accordingly, we chose the [θ]$_{max}$ value of the positive peak at 220–230 nm of X-PP5 to compare the PPII promotion of the X residues (Fig. 4e). While the introduction of various residues may occasionally cause a slight shift of 1–5 nm in the [θ]$_{max}$ wavelength[42–44,46], this measurement has been widely utilized in the PPII characterization of numerous peptides[40,42,43,45]. First, we verified that Hyp and Flp both have a strong PPII propensity with their [θ]$_{max}$ values higher than Pro (Hyp: 2655, Flp: 2532, Pro: 1948). These data validated the well-established conclusion that Flp and Hyp promote the pre-organization of single CMP chains into PPII, thus stabilizing the triple-helix[13,22]. Both Flp and Hyp have a fairly strong trans-amide propensity ($K_{cis/trans}$: 0.15 ~ 0.16)[21]. Second, the [θ]$_{max}$ values for the weakly trans-promoting N-glys ($K_{cis/trans} < 1$) were all notably greater than Pro, most of which were even higher than Hyp (Fig. 4e). Third, importantly, the [θ]$_{max}$ values of X-PP5 for the strongly cis-demanding

N-glys ($K_{cis/trans} > 1$) were still all higher than Pro, similar to or greater than Hyp and Flp, and overall slightly lower than the weakly trans-favoring N-gly group (Fig. 4e). These results, combined with our DSC and MD simulation data, provided solid evidence that peptoid residues, even the strongly cis-enforcing ones, can facilitate the pre-organization of individual peptide chains into the PPII conformation within a polyproline sequence, which may lead to the formation of an entropically favored triple-helix.

### Effect of the N-gly's cis-amide on CMP folding kinetics

Since N-glys' cis-trans propensities showed little effect on triple-helix stability, we wondered if they have an impact on the CMP folding kinetics. We first monitored the assembly of the X-CMP triple-helices from single-strands during a 0.5 °C min$^{-1}$ cooling process from 80 °C using CD at 225 nm (see Supplementary Methods). In our test, the X-CMPs for the amino acid guest residues and their N-gly counterparts ($K_{cis/trans} < 1$) all showed clear refolding, indicated by the rising CD signals when the temperature dropped below the corresponding $T_m$ values (Fig. 5a, b). In contrast, the CD curves of X-CMPs with the cis-demanding N-gly residues were linear throughout the cooling process (Fig. 5c) suggesting the lack of triple-helix folding. However, when we repeated the measurements for N2pic-CMP, Nme$_2$ae-CMP, Net$_2$ae-CMP, and N$^i$pr$_2$ae-CMP at higher pH levels to reduce the N-glys' cis-amide propensity by deprotonating their sidechains (Fig. 3), the four X-CMPs started to fold into the triple-helix during cooling (Fig. 5d, blue curves). Additionally, the CD cooling curves of Lys-CMP in PBS and NaOH were almost overlapping (Fig. 5d), indicating that simple removal of the positive charge from the sidechain of the X-residue has little effect on the triple-helix folding. These results suggested that the folding behavior of N2pic-CMP, Nme$_2$ae-CMP, Net$_2$ae-CMP, and N$^i$pr$_2$ae-CMP in the deprotonated state is more likely due to the reduction of the cis-amide propensity of their N-gly residues.

Next, we conducted the CD refolding kinetic studies of all the X-CMPs at 4 °C after heat-dissociating their triple-helices at 85 °C (see Supplementary Methods) and found similar trends. At the peptide concentration in our tests, the triple-helix folding of the X-CMPs follows a third-order kinetic model[47], allowing us to calculate the rate constant ($k_3$) for each X-CMP (see Supplementary Methods and Supplementary Fig. 5). For the amino acid guest residues, the CD signals of the X-CMPs increased rapidly toward considerable refolding at 120 min (Fig. 5e); for the weakly trans-biased X-peptoid

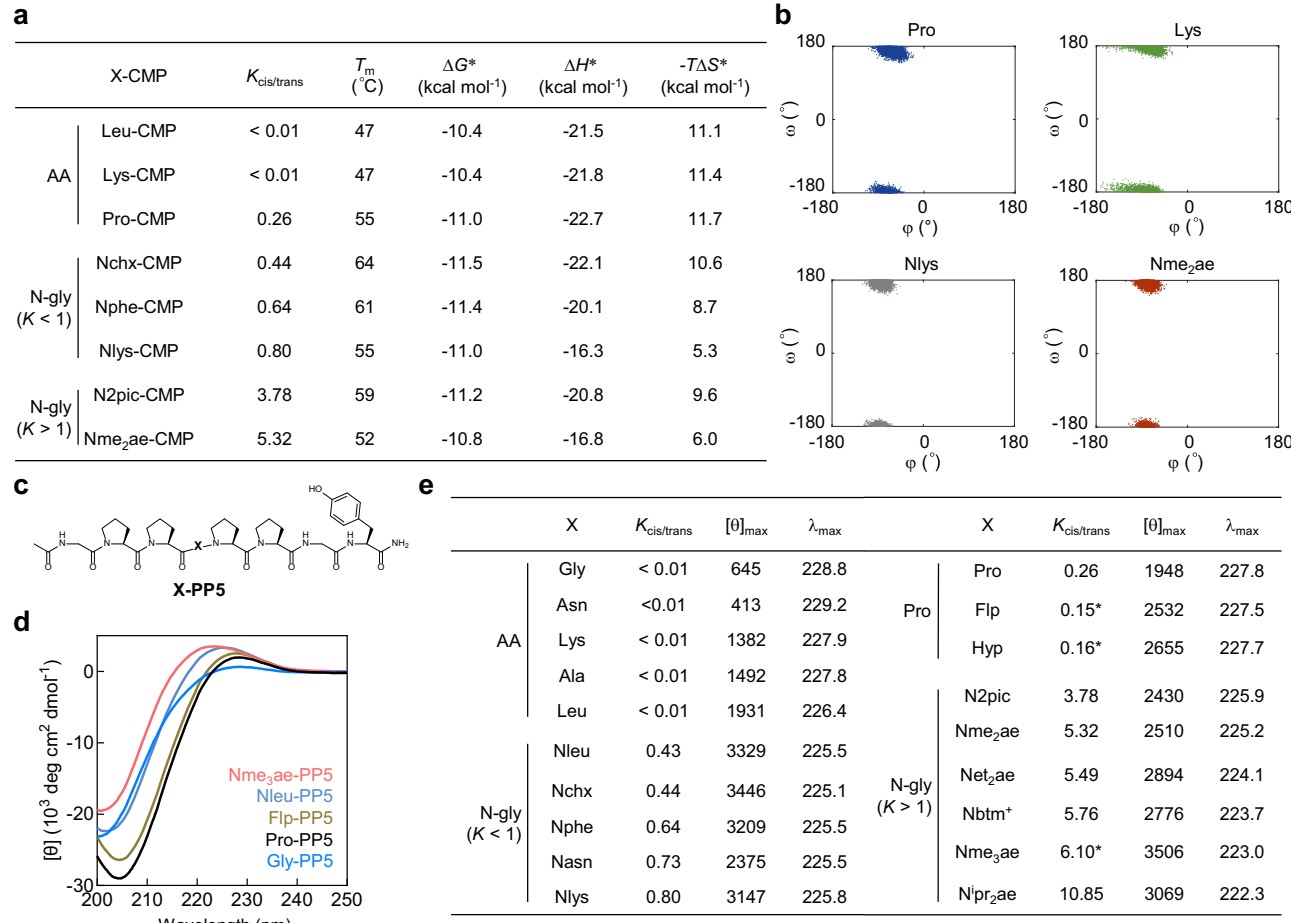

**a**

| | X-CMP | $K_{cis/trans}$ | $T_m$ (°C) | $\Delta G^*$ (kcal mol⁻¹) | $\Delta H^*$ (kcal mol⁻¹) | $-T\Delta S^*$ (kcal mol⁻¹) |
|---|---|---|---|---|---|---|
| AA | Leu-CMP | < 0.01 | 47 | -10.4 | -21.5 | 11.1 |
| | Lys-CMP | < 0.01 | 47 | -10.4 | -21.8 | 11.4 |
| | Pro-CMP | 0.26 | 55 | -11.0 | -22.7 | 11.7 |
| N-gly ($K < 1$) | Nchx-CMP | 0.44 | 64 | -11.5 | -22.1 | 10.6 |
| | Nphe-CMP | 0.64 | 61 | -11.4 | -20.1 | 8.7 |
| | Nlys-CMP | 0.80 | 55 | -11.0 | -16.3 | 5.3 |
| N-gly ($K > 1$) | N2pic-CMP | 3.78 | 59 | -11.2 | -20.8 | 9.6 |
| | Nme₂ae-CMP | 5.32 | 52 | -10.8 | -16.8 | 6.0 |

**e**

| | X | $K_{cis/trans}$ | $[\theta]_{max}$ | $\lambda_{max}$ | | X | $K_{cis/trans}$ | $[\theta]_{max}$ | $\lambda_{max}$ |
|---|---|---|---|---|---|---|---|---|---|
| AA | Gly | < 0.01 | 645 | 228.8 | Pro | Pro | 0.26 | 1948 | 227.8 |
| | Asn | <0.01 | 413 | 229.2 | | Flp | 0.15* | 2532 | 227.5 |
| | Lys | < 0.01 | 1382 | 227.9 | | Hyp | 0.16* | 2655 | 227.7 |
| | Ala | < 0.01 | 1492 | 227.8 | N-gly ($K > 1$) | N2pic | 3.78 | 2430 | 225.9 |
| | Leu | < 0.01 | 1931 | 226.4 | | Nme₂ae | 5.32 | 2510 | 225.2 |
| N-gly ($K < 1$) | Nleu | 0.43 | 3329 | 225.5 | | Net₂ae | 5.49 | 2894 | 224.1 |
| | Nchx | 0.44 | 3446 | 225.1 | | Nbtm⁺ | 5.76 | 2776 | 223.7 |
| | Nphe | 0.64 | 3209 | 225.5 | | Nme₃ae | 6.10* | 3506 | 223.0 |
| | Nasn | 0.73 | 2375 | 225.5 | | Nⁱpr₂ae | 10.85 | 3069 | 222.3 |
| | Nlys | 0.80 | 3147 | 225.8 | | | | | |

**Fig. 4 | Despite their greater cis-amide preference, X-peptoid residues reduce the entropy cost of X-CMP triple-helix folding compared to Pro, probably due to their overall stronger polyproline-II helix (PPII) propensity.**
**a** Thermodynamic parameters $\Delta H$, $T\Delta S$, and $\Delta G$ of X-CMP triple-helix formation were determined in PBS solution (except for N2pic-CMP, which was in 1 mM HCl). $T_m$ values were measured by CD. *: values were estimated by DSC and reported here at $T = 55\,°C$. **b** In the molecular dynamics simulation of the X-CMPs, the distributions of the (φ, ω) dihedral angles of the guest peptoid residues Nlys and Nme₂ae were similar to Pro, whereas the amino acid residue Lys had the greatest deviation in φ. **c** The structure of the PPII host-guest peptide X-PP5. **d** The CD spectra of Gly-PP5, Pro-PP5, Flp-PP5, Nleu-PP5, and Nme₃ae-PP5 showing the PPII characteristic positive band at 220–230 nm and negative band at 200–210 nm at 25 °C in 5 mM phosphate buffer (pH 7.0). **e** The $K_{cis/trans}$ values of the X residues with the $[\theta]_{max}$ (unit: deg cm² dmol⁻¹) and $\lambda_{max}$ (unit: nm) values of the corresponding X-PP5 peptides. *: values cited from ref. 21,31.

residues, the rise of the CD signal was slightly slower but generally comparable to their amino acid counterparts (Fig. 5e, f). The $k_3$ rate constant values were also generally comparable between the two groups (Fig. 5e, f). Similar to the results in Fig. 5c, the X-CMPs containing the strongly cis-biased peptoid residues displayed almost horizontal CD curves with minimal rate constants in the protonated state (Fig. 5g), but exhibited 4–8 times faster refolding after deprotonation of the N-gly sidechains (Fig. 5h). However, such pH- or protonation-responsive changes in refolding rate were not seen for Lys-CMP and other N-glys whose cis-trans isomerization is not affected by sidechain protonation (e.g., Nleu-, Nchx-, Nphe-CMPs, Supplementary Fig. 6). Altogether, these data demonstrated that the triple-helix folding rates of the trans-favoring X-N-glys can be comparable to the amino acids, whereas the strongly cis-inducing ones (with $K_{cis/trans}$ values substantially above 1) can drastically slow down the refolding by over an order of magnitude compared to Pro.

**Controlling the triple-helix folding by amide isomerization**
So far, our investigations demonstrated that the cis-amide propensities of the X-peptoid residues have little impact on the triple-helix stability (Figs. 2–4), but they can strongly hinder and delay the

formation of the CMP trimers (Fig. 5). In light of this discrepancy, we wondered whether and how one can control the triple-helix folding of a CMP by taking advantage of the cis-trans isomerization of its peptoid residues. This challenge is not only scientifically intriguing but also has value in practical applications. Previous research by us and others has demonstrated that, unlike the triple-helical CMPs that usually serve as a chemical model for native collagen structures, CMP single-strands can target the denatured collagen molecules in disease: a CMP strand may form a hybrid triple-helix with the denatured chains of natural collagen but not the intact triple-helical collagen proteins; this structural recognition has enabled detection of collagen denaturation due to proteolytic remodeling or mechanical damage in histopathology and in vivo for a myriad of conditions including lung fibrosis, intervertebral disc degeneration, and tendon failure[17,48]. However, conventional GPO-repeating CMP sequences will spontaneously form inactive peptide homotrimers over time, which must be preheated to dissociate into single strands before collagen hybridization (Supplementary Fig. 7). This preheating step has been hindering the biomedical development of collagen hybridization, especially in vivo applications[17]. To date, most synthetic strategies to retain a CMP in the single-strand state involve modifications to introduce repulsions

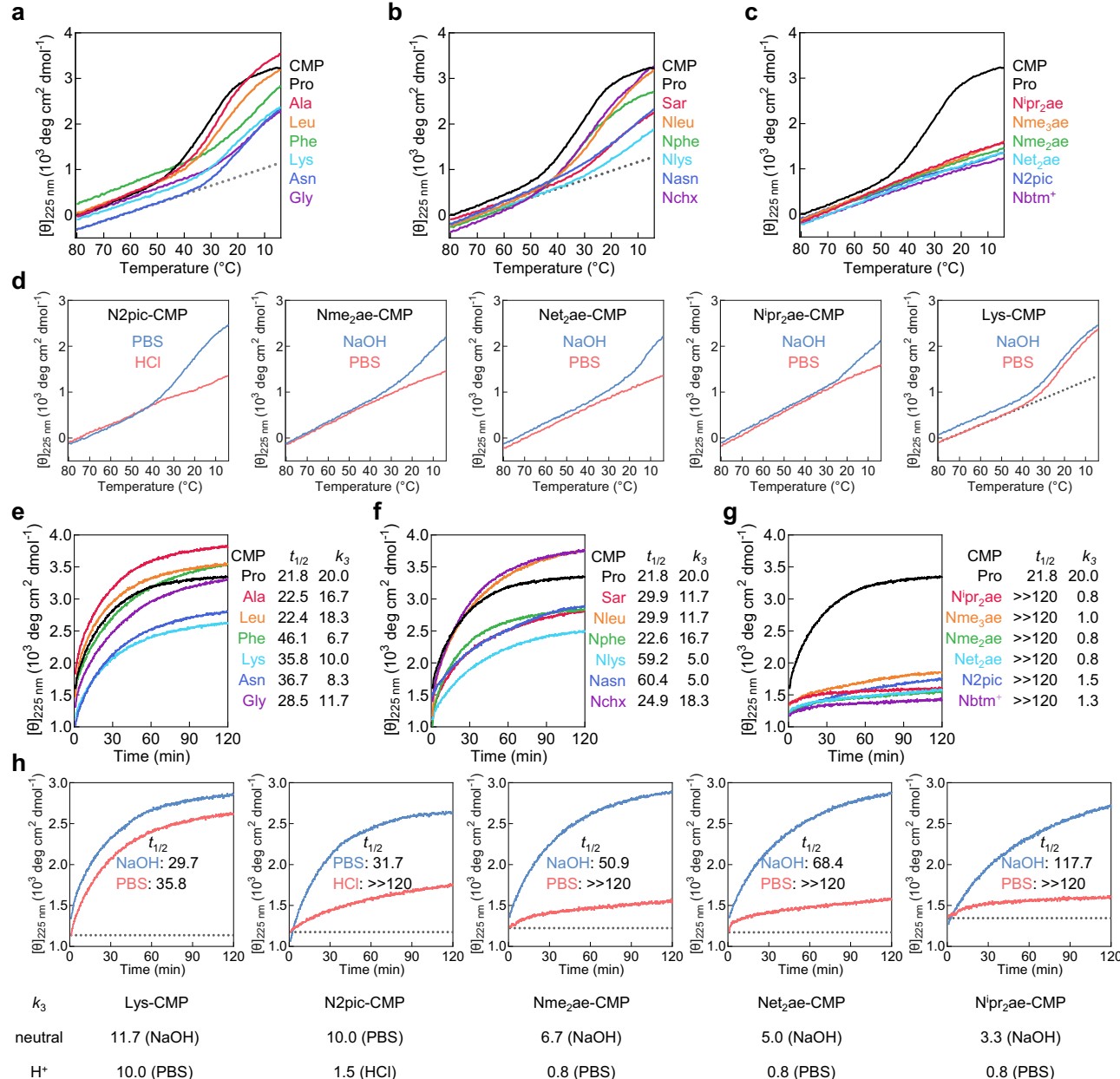

**Fig. 5 | Kinetic studies of the X-CMPs. a–d** CD cooling curves showing triple-helix folding of X-CMPs featuring amino acids (**a**), weakly trans-biased N-glys (**b**), and strongly cis-demanding N-glys (**c**) in PBS solution (except for N2pic-CMP, whose folding was measured in 1 mM HCl solution, pH 3.0), as well as N2pic-CMP, Nme₂ae-CMP, Net₂ae-CMP, Nⁱpr₂ae-CMP, and Lys-CMP in the protonated state (**d**, red) and the neutral state (**d**, blue) with a cooling rate of 0.5 °C min⁻¹ from 80 °C to 4 °C. **e–h** CD curves showing triple-helix refolding of X-CMPs featuring amino acids (**e**), weakly trans-biased N-glys (**f**), and strongly cis-demanding N-glys (**g**) in PBS (except for N2pic-CMP, whose refolding was measured in 1 mM HCl solution), as well as N2pic-CMP, Nme₂ae-CMP, Net₂ae-CMP, Nⁱpr₂ae-CMP, and Lys-CMP in the protonated state (**h**, red) and the neutral state (**h**, blue) at 4 °C in 120 min. The $t_{1/2}$ time (unit: min) was defined as the time at which 50% of the total ellipticity (measured before heating) was recovered. The third-order rate constants ($k_3$, unit: × 10³ M⁻² s⁻¹) were determined from the slopes of the fitted lines of the kinetic curves in Supplementary Fig. 5. All peptide solutions were heated at 85 °C for 10 min immediately before CD monitoring (**e–h**).

| $k_3$ | Lys-CMP | N2pic-CMP | Nme₂ae-CMP | Net₂ae-CMP | Nⁱpr₂ae-CMP |
|---|---|---|---|---|---|
| neutral | 11.7 (NaOH) | 10.0 (PBS) | 6.7 (NaOH) | 5.0 (NaOH) | 3.3 (NaOH) |
| H⁺ | 10.0 (PBS) | 1.5 (HCl) | 0.8 (PBS) | 0.8 (PBS) | 0.8 (PBS) |

between the peptide strands at the cost of the peptide's triple-helix folding and collagen-hybridizing capacity[17,49,50]. As such, we envisioned that the cis-demanding peptoid residues may provide a unique opportunity to resolve this dilemma (Fig. 6a): a cis-amide-containing X-CMP with a strong triple-helix propensity may be kept as single-strands in storage until direct in vivo administration to trigger their trans-amide folding in the physiological condition for targeting denatured collagen with high affinity.

Therefore, we designed and synthesized N2pic3-CMP, featuring three consecutive Gly-N2pic-Hyp triplets within the CMP host sequence (Fig. 6b). As a strongly cis-favoring peptide residue, N2pic

has a high $K_{cis/trans}$ value of 3.78 in acidic pH when its pyridine sidechain (pKa: 5.2) is protonated, but it becomes weakly trans-favoring in PBS buffer ($K_{cis/trans}$ = 0.72, Fig. 3b). As we expected and showed in Fig. 6b, N2pic3-CMP remained single-stranded in 1 mM HCl (pH 3.0) even after weeks of incubation at 4 °C (Supplementary Fig. 8), but formed a triple-helix ($T_m$: 69 °C) substantially more stable than the Pro-CMP host ($T_m$: 55 °C) in PBS buffer (pH 7.4). The flat refolding CD curve at pH 3 revealed that the three cis-demanding N2pic residues could essentially bring the CMP's triple-helix folding to a halt (Fig. 6c), yet it can be regained robustly at pH 7.4 (Fig. 6c). Moreover, we demonstrated that the triple-helix folding of N2pic3-CMP could be controlled on demand

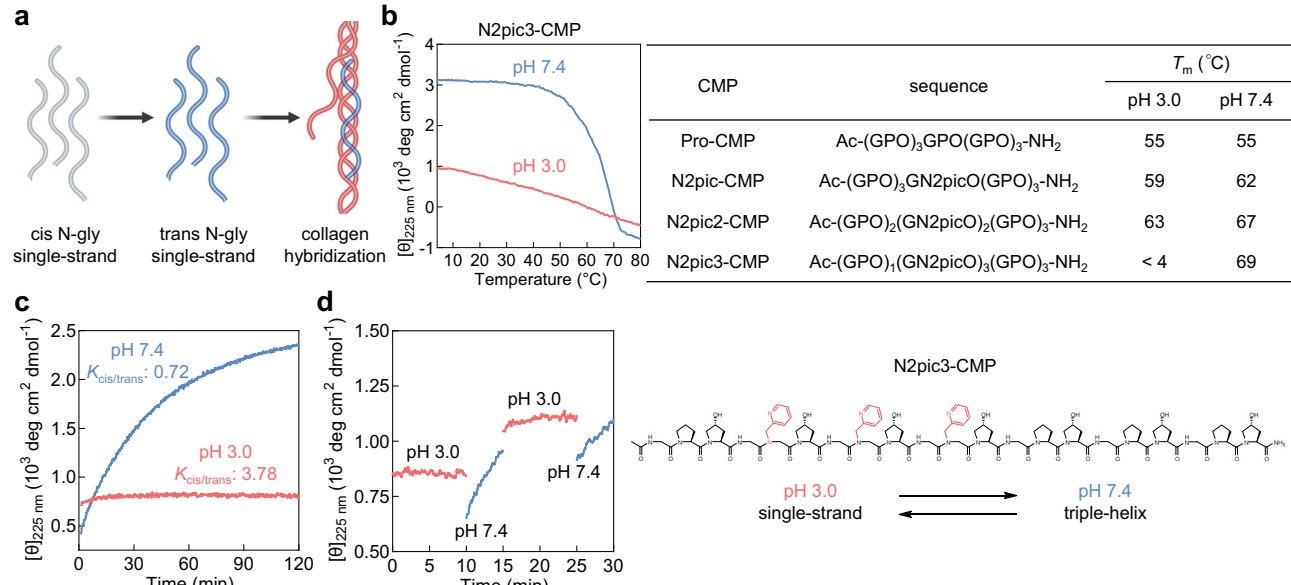

**Fig. 6 | Controlling the triple-helix folding of N2pic3-CMP by cis-trans iso-merization. a** A schematic showing that the triple-helix folding of a CMP can be triggered by switching the cis-trans isomerization of its peptoid residues for hybridization with denatured collagen chains. **b** The CD thermal unfolding curves showing that N2pic3-CMP remained single-stranded at pH 3.0 (1 mM HCl) but formed a triple-helix more stable than Pro-CMP at pH 7.4 (PBS). **c** The CD refolding curves of N2pic3-CMP at pH 3.0 (1 mM HCl, red) and 7.4 (PBS, blue). **d** The CD time-course curves showing that the refolding of N2pic3-CMP can be repeatedly trig-gered at pH 7.4 and stopped at pH 3.0. The pH levels were adjusted by alternately pipetting 1 μL of 0.23 M NaOH and HCl into the sample cuvette.

by changing the pH of the solution to switch the cis-trans isomeriza-tion of the N2pic residues (Fig. 6d): the triple-helix folding of N2pic3-CMP started immediately after we raised the pH of the solution to the physiological level and stopped as soon as the pH fell to 3.0 (see Supplementary Methods).

**Targeting denatured collagen in myocardial infarct in vivo**

We labeled N2pic3-CMP and the control peptides with a Cy5 fluor-ophore to test their ability to hybridize with heat-denatured collagen (i.e., gelatin) in vitro. As expected, the host peptide Cy5-Pro-CMP did not show binding unless it was heat-dissociated to single strands at 85 °C before being applied to the gelatin substrate (Fig. 7a). To avoid the preheating, we kept Cy5-N2pic3-CMP as single-strands in a dilute HCl solution (1 mM, pH 3.0) and directly mixed it into a PBS buffer sitting on the gelatin substrate, which neutralized the acid and induced the collagen hybridization. As anticipated, Cy5-N2pic3-CMP bound to the gelatin directly and exhibited a higher affinity than Cy5-Pro-CMP (Fig. 7a), a result correlating with their triple-helix stabilities (Fig. 6b). In addition, with a flipped GXY-repeating sequence, the control pep-tide Cy5-N2pic3-sCMP [sequence: (GlyProHyp)-(N2picHypGly)₃-(GlyProHyp)₃] completely lost the capacity to hybridize with gelatin (Fig. 7a). As we expected (Fig. 6a), Cy5-N2pic3-CMP stored in dilute HCl maintained its single-strand state, enabling direct intravenous injec-tion without preheating and in vivo targeting of the physiological level of denatured collagen in the spine and joints of normal mice (Sup-plementary Fig. 9).

Finally, we challenged the capacity of N2pic3-CMP to target the denatured collagen in the fibrotic scar of myocardial infarction (MI) in vivo. Following MI, massive collagen remodeling occurs, involving drastic collagen production and simultaneous degradation of the col-lagen matrix mediated by the infiltrating leukocytes, resulting in extensive collagen damage and denaturation[51,52]. One hour following intravenous injection into the MI mice whose left anterior descending arteries had been occluded for 7–14 days, the Cy5-N2pic3-CMP single-strands showed robust fluorescence signal in the infarcted myo-cardium, but minimal signals in the normal hearts from the sham group (Fig. 7b). Meanwhile, the control peptide Cy5-N2pic3-sCMP showed

barely any uptake in the infarcted hearts (Fig. 7b). Histologically, after collecting and cryo-sectioning the MI hearts with the in vivo-administered Cy5-N2pic3-CMP bound (see Supplementary Methods) we found that the CMP fluorescence was predominantly located within the infarcted and fibrotic areas (Fig. 7c, d) and colocalized with signals from a collagen antibody counter-stain (Fig. 7d). Light sheet fluores-cence microscopy of a cleared (see Supplementary Methods)[53] whole heart harvested from another MI mouse dosed with Cy5-N2pic3-CMP in vivo, confirmed our findings three-dimensionally (Fig. 7e and Sup-plementary Movie 1). Additionally, we synthesized a Pro-free CMP, where each of its seven X positions is occupied by a peptoid residue, and also validated its targeting to the denatured collagen in myocardial infarct in vivo (Supplementary Fig. 10). Overall, these results showcased that peptoid residues can enable creative designs of collagen peptido-mimetics with not only a strong capacity to target denatured collagen but also controllable folding for desired functional applications.

## Discussion

Previous studies on the effect of the peptide bond cis-trans iso-merization on protein structures (including collagen) had to rely on proline and modified prolines[11–13,18,54]. Because they only span a nar-row range of $K_{cis/trans}$ ratios (e.g., 0.14−0.45)[13], fresh insights into the structural effects of the cis-preferring amides on collagen (and probably other proteins as well) have been limited. However, as early as 1996, Goodman and co-workers showed that replacing Pro with some peptoid residues such as Nleu within CMPs results in triple-helix stability[55], which they attributed to the hydrophobic interchain interactions between the N-gly's sidechain and the adjacent Pro[55,56]. As a result, perhaps, the effect of the peptoid residue's cis-trans propensity on collagen triple-helicity has never been investigated. Meanwhile, the thermodynamics[31] and kinetics[57] of the cis-trans iso-merization have been characterized both experimentally and com-putationally for a variety of monomers in peptoids, many of which exhibit a robust cis-amide propensity[23,58]. Using these unnatural peptoid residues, our study interrogated the residues with a real cis-amide propensity ($K_{cis/trans} > 1$) and expanded the range of this ratio by an order of magnitude from the proline derivatives [0.26 (Pro)

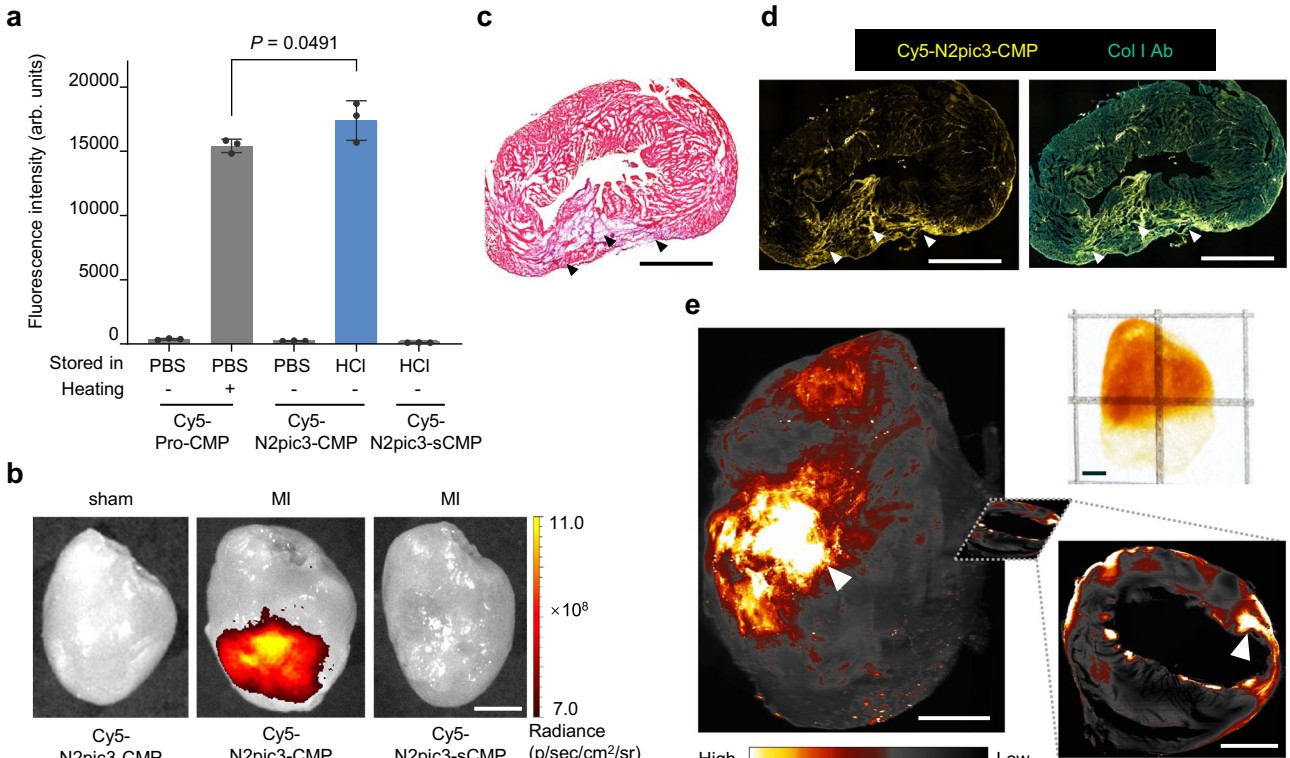

**Fig. 7 | In vivo targeting of denatured collagen in the fibrotic scar of myocardial infarction (MI) with N2pic3-CMP without preheating. a** Fluorescence of gelatin films treated with Cy5-labeled X-CMPs ($n = 3$ independent samples) demonstrating that Cy5-N2pic3-CMP stored in 1 mM HCl could directly hybridize with denatured collagen in PBS without preheating and had a higher affinity than Cy5-Pro-CMP. This collagen hybridization was abolished for the control peptide Cy5-N2pic3-sCMP with a flipped sequence [GlyProHyp-(N2picHypGly)$_3$-(GlyProHyp)$_3$]. Data were represented as mean ± standard deviation and analyzed with one-way analysis of variance (ANOVA), followed by post-hoc Tukey tests. **b** Representative near-infrared fluorescence images of the hearts harvested from mice 10 days after MI showed a robust fluorescence signal compared to the sham group, 1 h post

intravenous injection of Cy5-N2pic3-CMP, but not Cy5-N2pic3-sCMP ($n = 3$ mice). **c** Hematoxylin and eosin staining of the slides of the MI heart sectioned from (**b**) indicated the presence of infarction (arrowheads), $n = 3$ mice. **d** Fluorescence micrographs of the cryosections of the MI heart from the mouse injected with Cy5-N2pic3-CMP in vivo in (**b**), co-stained with an anti-collagen antibody, showing specific binding of Cy5-N2pic3-CMP to denatured collagen co-localizing with the collagen antibody within the infarcted fibrotic areas (arrowheads), $n = 3$ mice. **e** A photograph (top right) and light sheet microscopy fluorescence images of an MI heart after tissue clearing (from another mouse intravenously injected with Cy5-N2pic3-CMP) displayed massive collagen denaturation within the MI fibrotic scar (arrowheads), $n = 2$ mice. Scale bars: 2.5 mm (**b**), 2 mm (**c**, **d**) and 1.5 mm (**e**).

$< K_{cis/trans} < 10.85$ (N$^i$pr$_2$ae)] in collagen. We aim to bridge the gap between proline and peptoid residues in protein folding by investigating the amide cis-trans isomerization in collagen.

We found that the cis-trans propensity of a peptoid residue does not have a dominant effect on the stability of a collagen triple-helix (Figs. 2 and 3, see Supplementary Fig. 11 for statistics). This seems to contradict the fact that all peptide bonds in the collagen triple-helix must be trans. Nonetheless, our findings demonstrate that this discrepancy can be explained by thermodynamics: the stabilization is dominated overall by the peptoid residues' strong PPII folding, which overcomes their cis-amide propensity. (1) Our DSC data indicated that almost all X-peptoid residues offer a more favorable entropy term for the triple-helix folding than Pro (Fig. 4a), despite their stronger cis-amide preference. This result directly supports that the peptoid residues, even the cis-enforcing ones, can facilitate the conformational pre-organization of the triple-helix[35]. (2) The same X-peptoid residue can form triple-helices with almost identical stabilities despite its opposite trans-cis preferences at different pH levels, (e.g., N2pic-CMP and Net$_2$ae-CMP, Fig. 3). These data suggest that thermodynamically the cis and trans-isomers of a peptoid residue can contribute similarly to the triple-helix formation. Indeed, the trans isomer of an imide bond was calculated to be favored over the cis by 0.5 kcal mol$^{-1}$ in Pro-containing peptides[59]. This energy cost could be compensated by the large entropy benefit from a Pro→N-gly replacement in X-CMP (Fig. 4a). (3) Every N-gly residue we tested in

X-PP5 offered a signature PPII CD signal notably higher than Pro (Fig. 4e), including N$^i$pr$_2$ae with a $K_{cis/trans}$ value as great as 10.85. These data support that within a Pro-rich sequence, the cis-favoring peptoid residues can not only accommodate the trans-amide configuration but also promote the PPII folding robustly. Such delicate energetic interplay of local conformation preferences and the thermodynamic stability of the assembled structures has been well demonstrated in the studies of peptoid nanosheets[58,60]. Altogether, our data corroborate that despite their stronger-than-Pro cis-amide preference, the peptoid residues' overall high PPII propensity may help pre-organize the CMP single-chains, leading to lower entropy costs for the triple-helix stabilization.

We also found that while the ratio between the cis- and trans-isomers of Ac-X-OMe can generally reflect the cis-amide propensity of the X residue, its actual selectivity in the peptide-bond configuration in proteins may not be as strong as it appears in monomers, especially in terms of thermodynamics (Figs. 2–4). As previously reported, although the trans-amide was found to be favored over the cis by 2.5 kcal mol$^{-1}$ for the small-molecule peptide-bond analog N-methylacetamide[5,61], the difference for an imide bond was calculated to be as low as 0.5 kcal mol$^{-1}$ within Pro-containing peptides[59]. This discrepancy may be due to the involvement of multiple structural factors in peptides and proteins (e.g., backbone folding, sidechains) other than the peptide-bond configuration. Because the Ac-X-OMe type of small-molecule models has been widely employed for

evaluating the role of the X-residues' cis-trans propensity in peptide and peptoid folding[23,30], including CMPs[12,13,21,54], our findings caution that these monomer models may not fully reflect the steric and electronic contributions of the neighboring residues. Therefore, other assessments, such as those using a trimer (e.g., Gly-X-Hyp) or an oligomer model (e.g., X-PP5), may be desired to objectively determine the overall folding propensity of an X-unit in action, especially for the peptoid residues. Future systematical investigations are required to define the more representative models.

Previous studies have shown that proline derivatives with an electron-withdrawing group, such as Flp, Hyp, and 4(R)-azidoproline, are biased toward the *exo* proline pucker and can also enhance the proline's trans-amide propensity through an $n \rightarrow \pi^*$ interaction: imposing a $C^\gamma$-*exo* pucker on a pyrrolidine ring of a CMP pre-organizes not only the φ, ψ angles for triple-helix formation but also the ω angle[13]. Such interconnected preferences for the *exo* pucker and the trans-amide synergistically stabilize the triple-helix, yet making it difficult to examine the effect of individual factors[18,22,62]. The cis-preference of the peptoid residues in our study is less affected by the φ and ψ angles, but primarily driven by the steric effect of the bulky sidechain (e.g., N-gly: Pro $< K_{cis/trans} < 1$) as well as the interactions between the sidechains and the backbone carbonyl (e.g., N-gly: $K_{cis/trans} > 1$). Therefore, we were able to directly interrogate the impact of the cis-trans preference with the influence from the φ, ψ angles largely disengaged. For the CMPs featuring modified prolines (e.g., fluoroproline, aminoproline)[22,28], the ω angle is a key variable for triple-helix formation, probably because the φ, ψ angles are greatly constrained by the pyrrolidine ring. However, for the peptoid residues, when the φ, ψ angles are no longer covalently restricted, the cis-trans isomerization may become less significant. Indeed, all other natural amino acids have a higher trans-amide propensity than proline, but without the suitable φ, ψ angles imposed by the proline ring, none of them can stabilize the triple-helix as effectively as Pro (Figs. 2 and 4a)[63].

Current understandings of the impact of the cis/trans-preference on the triple-helix stability largely focus on the Yaa position[12,18,22], while a few studies have indicated opposing structural effects from the same residue at different positions. (1) For example, the strongly trans-amide favoring Flp can greatly stabilize the triple-helix at the Yaa position but disrupts the structure at the Xaa position [e.g., (ProFlpGly)$_7$, $T_m = 45$ °C; (FlpProGly)$_7$, no triple-helix][21,22]. This was attributed to the mismatch between Flp's preference for the *exo* pucker and the favored *endo* pucker for the Xaa location[22]. (2) Moreover, the *endo*-pucker preferring flp can form a more stable triple-helix than Pro at the Xaa location [e.g., (flpProGly)$_7$, $T_m = 33$ °C; (ProProGly)$_7$, no triple-helix] even with its reduced trans-amide preference ($K_{trans/cis}$: flp vs. Pro = 2.5 vs. 4.6)[22,26]. These results from modified prolines suggested that proper φ, ψ dihedral angles may be more important for the CMP triple-helix than the ω angle at the Xaa position[22,64], and nicely corroborated our findings from the peptoid residues in this study. Importantly, while we believe our conclusions apply to the Xaa position, further investigation is required to determine how the cis-trans preference of the peptoid residues affects the triple-helix stability at the Yaa position.

The cis-trans isomerization of proline and its analogs is often key to switching protein conformations and controlling ligand binding[1–3,65]. In this study, we have proved the concept of controlling the collagen triple-helix folding by switching the cis-trans isomerization of a peptoid residue (Fig. 6). The halted refolding for the cis-favoring peptoid residues (Fig. 5c, g) in comparison to the normal refolding for the weakly trans-favoring ones (Fig. 5b, f), as well as the opposite folding behaviors of the same X-CMPs with the difference in amide preference of a single residue (Fig. 5d), all suggested that the triple-helix folding may only proceed when the peptoid amide bond is in the trans configuration. Overall, the cis-trans isomerization of the peptoid monomer can direct the X-CMP

triple-helix folding as a rate-limiting step in the sense that the folding is promoted when $K_{cis/trans} < 1$ but clearly inhibited when $K_{cis/trans}$ is substantially above 1 (Supplementary Fig. 5). Apparently, the cis-demanding peptoid residues can vastly raise the kinetic activation energy barrier of the peptide assembly, but this energy barrier is lowered when the peptoid residues are deprotonated and transformed into trans-amide favoring (Fig. 5d, h). This kinetic property made it possible to turn the triple-helix formation of a CMP on and off simply by switching the cis-trans propensity of its peptoid residues (Fig. 6d). Moreover, because some peptoid residues have a higher triple-helix propensity than Pro, this approach enables the design of a non-self-trimerizing CMP single-strand (Supplementary Fig. 7) with a stronger capacity for triple-helical hybridization than the conventional Pro-CMP (Fig. 7a), as well as those non-self-trimerizing CMPs that rely on chain-repulsing modifications[49,50,66]. Here we chose N2pic, whose trans-amide isomerization and triple-helix folding can be triggered by adjusting the pH to the physiological level, and produced a CMP that is ideal for in vivo collagen hybridization with high affinity and no need for preheating (Fig. 7 and Supplementary Fig. 9).

Myocardial infarction is a leading cause of death and a critical threat to human health. Precise imaging of the fibrotic scar of myocardial infarction is challenging in clinical diagnosis, and there is currently no disease-modifying treatment for myocardial fibrosis[67,68]. Recent studies have demonstrated significant collagen remodeling and degradation in the area of myocardial infarction along with extensive collagen denaturation within the fibrotic lesion[48,52,69]. To this end, we achieved in vivo targeting of the infarcted fibrotic lesion through collagen hybridization, as well as three-dimensional fluorescence mapping of collagen remodeling in the infarcted heart (Fig. 7). Our results also validated that a CMP strand can maintain its collagen-hybridizing capacity in vivo even with all Pro residues replaced by peptoid units (Supplementary Fig. 10). As dozens of peptoid monomers can be utilized to create huge CMP libraries with a robust triple-helix propensity[25], we anticipate further development toward imaging contrast agents and bioactive peptidomimetics to target collagen denaturation and remodeling for the diagnosis, monitoring, and treatment of an array of fibrotic diseases.

## Methods

A detailed description of the materials and methods is provided in the Supplementary Methods.

### Solid-phase synthesis

All sequences were prepared on Rink Amide AM resin using the Fmoc strategy. Fmoc-deprotection was carried out by 20% piperidine. HATU and HOAt were used as the coupling reagents. Peptoid residues were incorporated on-resin by the sub-monomer approach using bromoacetic acid and N,N'-diisopropylcarbodiimide (DIC) before the desired amine sub-monomer was added[24]. All peptides were purified by HPLC and verified by MALDI-MS.

### CD and DSC studies

All peptide solutions (150 μM for CD or 250 μM for DSC) were incubated at 4 °C for at least 48 h before measurements. CD spectra were taken from 200 to 260 nm at 4 °C (for X-CMP) or 25 °C (for X-PP5). Thermal melting curves were obtained by monitoring the ellipticity of each peptide solution at 225 nm from 4 to 80 °C with a heating rate of 0.5 °C min$^{-1}$. To obtain the refolding curves, peptide solutions (150 μM) were preheated at 85 °C before being immediately cooled to 4 °C, after which their ellipticity at 225 nm was monitored for 120 min. DSC measurements were performed according to Shoulders et al.[34,35], from 20 to 80 °C with a scan rate of 0.5 °C min$^{-1}$. The corresponding reference scans were subtracted from the sample scans.

### In vivo targeting of myocardial infarction

All animal studies were approved by the experimental animal use and ethics committee of the Fifth Affiliated Hospital of Sun Yat-sen University (Project License: 00299). Mice were housed in the pathogen-free room with a temperature of 20–26 °C and a humidity of 40–70%, a 12-h light/12-h dark cycle. Briefly, a mouse was anesthetized and the heart was exposed by thoracotomy through the fourth intercostal space, and the left anterior descending artery (LAD artery) was permanently ligated with an 8-0 nylon suture. Mice in the sham group underwent the same surgical procedures as the MI group except for LAD ligation. Seven days after LAD ligation, 4 nmol of Cy5-N2pic3-CMP or Cy5-N2pic3-sCMP were directly injected into the tail vein of each mouse without preheating. The mice were sacrificed 1 h post-injection, and the hearts were harvested for imaging and histology.

### Reporting summary

Further information on research design is available in the Nature Portfolio Reporting Summary linked to this article.

## Data availability

The data generated in this study are provided in the Supplementary Information and the Source Data file. Any additional requests for information can be directed to, and will be fulfilled by, the corresponding authors. Source data are provided with this paper.

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

## Acknowledgements

This work was supported by the National Natural Science Foundation of China (92059104 and 82071977 to Y.L.), the 2018 High-level Health Team of Zhuhai (awarded to Y.L.), and the Guangdong-Hong Kong-Macao University Joint Laboratory of Interventional Medicine Foundation of Guangdong Province (2023LSYS001).

## Author contributions

R.Q., Z.Q., G.L. and Y.L. designed the study. R.Q., X.L., K.H., W.B., D.Z. and Z.Q. performed research and curated data. R.Q., X.L., G.L., Z.Q. and Y.L. analyzed the data. All authors contributed to the writing and editing of the manuscript.

## Competing interests

The authors declare no competing interests.
