## [Peer Review File · Nature Communications]

Reviewers' Comments:

Reviewer #1:

Remarks to the Author:

In this study, the authors generate a series of collagen mimetic host-guest peptides with N-substituted amino acids (specifically glycine) - also referred to as peptoids. These peptides and natural collagens contain proline and hydroxyproline which have a greater propensity to form cis-amide bonds. In the triple-helix folded form of collagen, all peptide bonds must be trans - hence it is proposed that by incorporating a series of N-substituted glycines with varying cis/trans propensities, it may be possible to alter their stability. The goal was to develop better reagents for targeting areas of fibrotic tissue in vivo where peptides hybridize to denatured collagen in situ. A current challenge of these imaging approaches is the competition between homotrimerization of the peptide and hybridization to the tissue. Some of the designs fold very slowly, especially at low pH, allowing them to be administered in the single stranded form. They can be used to image infarctions in vivo without requiring a preheating step to disrupt homotrimers.

Overall this study is a nice integration of fundamental chemistry and medical translation, and is of the significance and caliber for publication in this journal. My concerns were with some of the experimental and computational methodologies. Specifically:

(1) the authors note that the stabilities and cis/trans propensities do not strongly correlate. This is true based on the given data. However, the melting experiments were done at a fast rate (0.5 C / min). This is not a fast rate for most proteins, but as the authors note, collagen folding is quite slow and does not equilibrate on time scales typical for other proteins. Notably Persikov and Brodsky (Equilibrium thermal transitions of collagen model peptides - Protein Science 2004) showed that at rates above 0.1 C / min at typical concentrations for peptide experiments, most sequences will show the same transition temperature. The same is true for the calorimetry experiments. Slow denaturation experiments - sometimes overnight depending on the temperature range and peptide concentration - are the standard in the field. I am unsure that the weak correlation reported here is real - it may be an artifact of the rapid heating schedule.

(2) Molecular dynamics are used to study the conformational preferences of the various N-substituted peptides. Notably the forcefields do not incorporate N- π star interactions that contribute to cis-trans propensities. The resulting analysis is likely misleading.

Reviewer #2:

Remarks to the Author:

Trans amide bonds are almost found exclusively in peptides except for pro residue. Proline is known to adopt both trans and cis conformation in peptides, however, for collagen, the most prevalent proline-abundant protein, all peptide bonds must be trans to form its hallmark triple-helix structure. In this article, using host-guest collagen mimetic peptides (CMPs), authors discovered that cis enforcing peptoid residues (N-substituted glycines) form stable triple-helices. These peptoid residues entropically stabilize the triple-helix by preorganizing individual peptides into a polyproline-II helix. Following that, authors designed a CMP whose triple-helix formation can be controlled by peptoid cis-trans isomerization, enabling direct targeting of fibrotic remodeling in myocardial infarction in vivo. The control of such collagen mimetics formation allows programming future peptidomimetics. A few points are suggested below:

1. Authors have reported peptoid residue containing collagen previously (J. Am. Chem. Soc. 2021, 143, 29, 10910–10919). Can authors justify why the current discovery warrants publication in Nat Commun?
2. It is not clear how cis or trans conformation of amide bond was measured. It seems that it is determined by Ac-X-OMe ¹H NMR, however, when the residue is incorporated into collagen sequence, there is evidence to show that the peptoid residue would preserve conformation.
3. The full spectrum of CD spectrum for those peptoid containing collagen at certain temperature (e.g. 25°C) should be presented. Readers would be able to directly compare the intensity of the

CD signatures, from which the stability and conformation of collagen could be predicted. Then it is easier for readers to follow the temperature dependent stability based on CD in the article.

3. Many groups are working on collagen mimetics, such as Jeffrey D. Hartgerink at rice university, Michael Yu at Utah University, David M. Chenoweth at U Penn. Please cite their references in the introduction.

Reviewer #3:

Remarks to the Author:

In this article, the authors found that substitution of cis-inducing peptoid monomers in collagen mimetic peptides (CMPs) do not affect the stability of assembled triple helix structures, but they can significantly retard the folding rate. In the case of several cationic cis-inducing monomers, it was found that under acidic conditions (where the monomer is strongly cis-inducing) the folding rate was very slow, but could be significantly accelerated when base was added to deprotonate the side chain (where the monomer is no longer cis-inducing). This property allowed the authors to trigger the assembly of these analogs in the presence of damaged collagen in vivo to image fibrotic lesions caused by myocardial infarction. The work is carefully performed, and the topic is interesting to the general audience of Nature Communications, however the key points of this work are not clearly presented.

The kinetics and thermodynamics of cis-trans isomerization in peptoids is a well-studied property of peptoids, and prior experimental and computational work in this area is not adequately referenced and discussed. There is also history, dating back to Murray Goodman and others, on the substitution of peptoid residues and related analogs into collagen, and that work is also not adequately referenced or discussed. Much of the paper is devoted to the study of the cis/trans propensity of model peptoid monomers (the Ac-X-OMe model), and how this is not correlated to the melting temperature of the triple helix. This is a simple comparison to make, however, the monomer model system is not particularly appropriate since it does not reflect the steric and electronic contributions of neighboring residues, making this correlation – which is the basis for most of the paper – less impactful. A trimer model, like Gly-X-Hyp would be more representative. The fact that cis-induction does not destabilize the triple helix, but does impact the rate of folding is fascinating and could be more clearly rationalized and described in kinetic and thermodynamic terms earlier in the paper, with perhaps less emphasis on the monomer model system. For example, does the $K_{cis/trans}$ or the rate of cis-trans isomerization of the peptoid monomer correlate directly with the triple helix folding rate? In other words, does folding only proceed when the peptoid is in the cis conformation. Is amide isomerization rate limiting or is the cis CMP assembling into the triple helix at a slower rate without amide isomerization)? The delicate energetic interplay of local conformation preferences and the thermodynamic stability of folded/assembled structures is an excellent contribution of this work and should be put into context with other similar findings in the peptoid field [e.g. Whitlam, PNAS 115, 5647 (2018)].

Reviewer #1 (Remarks to the Author):

In this study, the authors generate a series of collagen-mimetic host-guest peptides with N-substituted amino acids (specifically glycine) - also referred to as peptoids. These peptides and natural collagens contain proline and hydroxyproline, which have a greater propensity to form cis-amide bonds. In the triple-helix folded form of collagen, all peptide bonds must be trans - hence, it is proposed that by incorporating a series of N-substituted glycines with varying cis/trans propensities, it may be possible to alter their stability. The goal was to develop better reagents for targeting areas of fibrotic tissue in vivo where peptides hybridize to denatured collagen in situ. A current challenge of these imaging approaches is the competition between homotrimerization of the peptide and hybridization to the tissue. Some of the designs fold very slowly, especially at low pH, allowing them to be administered in the single-stranded form. They can be used to image infarctions in vivo without requiring a preheating step to disrupt homotrimers. Overall, this study is a nice integration of fundamental chemistry and medical translation and is of the significance and caliber for publication in this journal. My concerns were with some of the experimental and computational methodologies. Specifically:

We thank the reviewer for the favorable comments and constructive suggestions. Please see below our point-by-point responses to individual reviewer comments (in blue).

1. The authors note that the stabilities and cis/trans propensities do not strongly correlate. This is true based on the given data. However, the melting experiments were done at a fast rate (0.5 °C / min). This is not a fast rate for most proteins, but as the authors note, collagen folding is quite slow and does not equilibrate on time scales typical for other proteins. Notably, Persikov and Brodsky (Equilibrium thermal transitions of collagen model peptides - Protein Science 2004) showed that at rates above 0.1 °C / min at typical concentrations for peptide experiments, most sequences will show the same transition temperature. The same is true for the calorimetry experiments. Slow denaturation experiments - sometimes overnight, depending on the temperature range and peptide concentration - are the standard in the field. I am unsure that the weak correlation reported here is real - it may be an artifact of the rapid heating schedule.

Response and revision:

We thank the reviewer for the question on this important issue. We agree with the reviewer that the folding of collagen triple-helix is slow, and the measured T_m values depend on the heating rate. However, we are confident with our experimental results and the conclusion that the stabilities and cis/trans propensities do not strongly correlate, based on the following new data and reasoning.

(1) As suggested, we measured the T_m values of a group of key X-CMPs by CD under the heating rate of 0.1 °C/min. As expected, the new T_m values acquired with slow heating are consistently lower than the ones obtained with the 0.5 °C/min heating rate (this study) by ~6 °C, but the difference in T_m remained the same between individual X-CMPs. Notably, the difference in T_m between Pro-CMP and each peptoid-featuring X-CMP remains virtually unchanged under different heating rates (i.e., $\Delta\Delta T_m = 0$). These data confirm that the high stability of these peptoid X-CMPs is not artifacts caused by our heating rate.

b

X-CMP	$K_{cis/trans}$	Heating rate: 0.5 °C/min		Heating rate: 0.1 °C/min		$\Delta\Delta T_m$ (°C) (0.5 vs. 0.1 °C/min)
		T_m (°C)	ΔT_m (°C) vs. Pro	T_m (°C)	ΔT_m (°C) vs. Pro	
Pro-CMP	0.26	55	0	49	0	0
Nleu-CMP	0.43	59	4	53	4	0
Nchx-CMP	0.44	64	9	58	9	0
Nphe-CMP	0.64	61	6	57	8	-2
Nlys-CMP	0.80	55	0	48	-1	1
N2pic-CMP	3.78	59	4	53	4	0
Nme ₂ ae-CMP	5.32	52	-3	46	-3	0
Nme ₃ ae-CMP	6.10	53	-2	47	-2	0

In addition, we re-measured the DSC curves of several key X-CMPs at a heating rate of 0.1 °C/min. Consistent with the CD results, the stability relationship among these X-CMPs remained the same with the slower heating rate, despite a general 5~6 °C decrease in T_m value compared to the ones obtained with the 0.5 °C/min heating rate.

X-CMP	$K_{cis/trans}$	Heating rate: 0.5 °C/min		Heating rate: 0.1 °C/min		$\Delta\Delta T_m$ (°C) (0.5 vs. 0.1 °C/min)
		T_m (°C)	ΔT_m (°C) vs. Pro	T_m (°C)	ΔT_m (°C) vs. Pro	
Pro-CMP	0.26	54	0	48	0	0
Nphe-CMP	0.64	62	8	55	7	1
Nlys-CMP	0.80	54	0	48	0	0
N2pic-CMP	3.78	58	4	52	4	0
Nme ₂ ae-CMP	5.32	52	-2	45	-3	1

Considering that other readers may have similar questions regarding the heating rate, we have included these new CD data obtained with the 0.1 °C/min heating rate as new figures (Supplementary Fig. 1c,d) in the Supplementary Information and mentioned them accordingly in the results.

(2) In this study, we chose the heating rate of 0.5 °C/min to allow the comparison of our results to many previous studies on CMPs, especially those reporting the effect of the cis-trans amide propensity of the proline derivatives. See below.

CD heating rate	Reference
0.5~0.6 °C/min	J. Am. Chem. Soc. 2021, 143, 5937-5942. J. Am. Chem. Soc. 2019, 141, 5607-5611. J. Am. Chem. Soc. 2020, 142, 2208-2212. Chem. Commun. 2017, 53, 11036-11039. Chemistry. 2020, 26, 5070-5074. Angew. Chem. Int. Ed. Engl. 2019, 58, 3143-3146. Angew. Chem. Int. Ed. Engl. 2014, 53, 10340-10344. Chem. Sci. 2022,13, 12567-12576. J. Am. Chem. Soc. 2021, 143, 10910-10919.
0.1~0.167 °C/min	J. Am. Chem. Soc. 2011, 133, 5432-5443. J. Am. Chem. Soc. 2017, 139, 9427-9430. Nat. Chem. 2016, 8, 1008-1014. Biopolymers. 2012, 98, 111-121. Biopolymers. 2011, 96, 4-13. FEBS. J. 2008, 275, 5830-5840. Protein Sci. 2004;13(4):893-902.

We chose a heating rate of 0.5 °C/min for the DSC experiments to be consistent with our CD experiments. Fast heating rates have been reported in several DSC studies of collagen proteins and mimetic peptides.

DSC heating rate	Sample	Reference
2.5 °C/min	(GPO) ₂ GPRGMP(GPO) ₄ , GYO(GPO) ₂ (GKOGPO) ₂ GPO GPHGPM	PNAS. 2022;119(40):e2209524119.
2.0 °C/min	(GPO) ₇ , (POG) ₇	Chem Sci. 2022;13(42):12567-12576.
1.0 °C/min	biosynthetic collagen; (POG) ₁₀ ; recombinant collagen	Nat Commun. 2022;13(1):6761. Process Biochemistry. 2022, 121, 26-34. J Biol Chem. 2014;289(8):4941-4951.
0.5 °C/min	(GPO) ₃ GXO(GPO) ₃	This study
0.25 °C/min	(ProMopGly) ₁₀ , (ProFlpGly) ₁₀	J Am Chem Soc. 2008;130(10):2952-2953.
0.167 °C/min	2(EOGPOG) ₅ -(PRG) ₁₀ ; 2(PKGPOG) ₅ -(EOG) ₁₀ ;	J Am Chem Soc. 2010, 132(10): 3242-3243. J Am Chem Soc. 2011;133(14):5432-5443.
0.1 °C/min	(hypProGly) ₁₅ , (mopProGly) ₁₅ ; (flpMepGly) ₇ , (mepFlpGly) ₇ ; (Pro-Hyp ^R -Gly) ₁₀ , (Pro-fPro ^R - Gly) ₁₀	J Am Chem Soc. 2010;132(31):10857-10865. PNAS. 2010;107(2):559-564. Biochemistry. 2005;44(16):6034-6042.

The relationship between a CMP's T_m value and the heating rate during its measurement has been systematically characterized by Berisio *et al.* in 2004. According to their study, although the T_m values are always measured higher with a faster heating rate, the stability difference between two CMPs will not change significantly with the heating rate (see below). Our CD and DSC data also supported this conclusion.

FIGURE 1 Dependence of melting temperatures of [(PPG)₁₀]₁₃ (red), [(POG)₁₀]₃ (green), and [(PPG)₁₀]_{3urch} (black) upon scanning speed. Estimates of T_m values, derived from extrapolation to scanning speed $0\text{ }^{\circ}\text{C h}^{-1}$ are 32.0 , 58.5 , 39.1°C for the three polypeptides, respectively. A concentration of the species of 3 mg mL^{-1} was used during the experiments.

from Berisio R et al. *Characterization of collagen-like heterotrimers: implications for triple-helix stability. Biopolymers.* 2004;73(6):682-688. doi:10.1002/bip.20017.

In summary, based on our new data and these findings from previous studies, we believe our CMP stability measurements are meaningful and reliable, and we hope the reviewer's concerns are resolved.

2. Molecular dynamics are used to study the conformational preferences of the various N-substituted peptides. Notably, the forcefields do not incorporate N-pi star interactions that contribute to cis-trans propensities. The resulting analysis is likely misleading.

Response and revision:

We thank the reviewer for this important question. We have revised the original CGenFF parameters by reducing the energy across ω torsion angles (*i.e.*, lowering each ω force constant from 2.50 to 2.15 kcal/mol). This modification effectively lowers the cis/trans barrier relative to proteins, according to Weiser *et al.*, who showed that peptoid conformations computed with the modified-CGenFF reproduce the quantum mechanical potential energy profiles.

(39) Weiser, L.J. & Santiso, E.E. A CGenFF-based force field for simulations of peptoids with both cis and trans peptide bonds. *J. Comput. Chem.* **40**, 1946-1956 (2019).

We have re-run the simulations using this modified-CGenFF for the X-CMPs featuring the peptoid residues. We found that the modified CGenFF does not affect the overall stability of the X-CMPs as the CMPs with the peptoid residue substitutions are stable within the 100 ns simulation. Importantly, our simulation results and conclusions are still valid, as the distributions of the dihedral angles (ϕ , ψ , ω) of the peptoid residues do not vary much with the change in the force field parameters. Quantitatively, the deviations of the ϕ angle even get slightly smaller for Nlys, Nme₂ae, and Nme₃ae with the modified force field.

The key results are shown below.

Based on our new data obtained with the modified-CGenFF, we believe our MD simulation results and conclusions are reliable, and we hope that the reviewer's concerns are resolved. We have replaced the original simulation data with these new ones in the manuscript (see our new Fig. 4b, Supplementary Fig. 4, and updated Supplementary Methods).

Reviewer #2 (Remarks to the Author):

Trans amide bonds are almost found exclusively in peptides except for pro residue. Proline is known to adopt both trans and cis conformation in peptides, however, for collagen, the most prevalent proline-abundant protein, all peptide bonds must be trans to form its hallmark triple-helix structure. In this article, using host-guest collagen mimetic peptides (CMPs), authors discovered that cis enforcing peptoid residues (N-substituted glycines) form stable triple-helices. These peptoid residues entropically stabilize the triple-helix by preorganizing individual peptides into a polyproline-II helix. Following that, authors designed a CMP whose triple-helix formation can be controlled by peptoid cis-trans isomerization, enabling direct targeting of fibrotic remodeling in myocardial infarction in vivo. **The control of such collagen mimetics formation allows programming future peptidomimetics.** A few points are suggested below:

1. Authors have reported peptoid residue containing collagen previously (J. Am. Chem. Soc. 2021, 143, 29, 10910–10919). Can authors justify why the current discovery warrants publication in Nat Commun?

Response:

We thank the reviewer for this critical question.

First, although we and our collaborators have reported peptoid-residue containing collagen (*J. Am. Chem. Soc.* 2021, 143, 29, 10910–10919), this work is not simply an extension of our previous study. The scientific topics of this study are new. In the previous paper, we reported a series of peptoid residues that can form stable collagen triple-helices but **did not interrogate the relationship between the triple-helix stability and the peptoid residues' cis-trans propensity**. In fact, we explicitly put the following remark in the previous paper, underlining the need for a separate study focusing on the role of a residue's cis-trans propensity.

“In addition to the ϕ , ψ dihedral angles, the ω angle (i.e., the *cis-trans* isomerization of the Pro amide bond) is a key parameter determining the backbone folding of a CMP: the higher the *trans:cis* ratio of the Pro amide bonds in a single strand, the more stable the resulting triple-helix.³ This rule is well accepted for CMPs based on Pro, Hyp, and modified prolines (e.g., fluoroproline, azidoproline),^{17,49} whose ϕ , ψ angles are largely restrained by the pyrrolidine ring. However, for peptoid residues, how the *trans:cis* ratio of the N-gly's amides affects the triple-helix stability now becomes a topic for further investigation. In the current study, although residues Sar, NEt, and Nphe all seem to favor the *trans*-amide less than Pro, they generally stabilize the triple-helix, suggesting that the effect of the *cis-trans* isomerization may be more complex for residues that do not contain a proline ring.”

In this study, we systematically investigated and established the correlation between the cis-trans propensity of the peptoid residue and the collagen triple-helix structure. Over the past 30 years, studies of the effect of the peptide cis/trans-isomerization on collagen and other protein structures have been mainly focused on the amino acids and proline derivatives, which only span a narrow range of cis-trans amide propensity (blue region, below).

Here, by replacing Pro with peptoid residues, our study expands the charted spectrum of a residue's cis-trans propensity by an order of magnitude in protein folding and interrogates the impact of an actual cis-amide-favoring residue for the first time in collagen (red region, above).

Innovation and contribution to the field:

(1) For collagen, the peptide bonds of its three chains must all be trans to form its characteristic triple-helix structure. This fact has prompted the development of collagen mimetics solely relying on trans-amide-favoring residues ($K_{cis/trans} < 0.5$) since the 1990s. Here, we discovered that, surprisingly, even the strongly cis-demanding peptoid residues ($K_{cis/trans} > 3$) can still form collagen triple-helices as stable as Pro, challenging the conventional idea that the cis-promoting residues are destabilizing for collagen. Thermodynamically, we provided vital DSC data and established that the stabilization comes from the peptoid residues' preorganization of individual collagen peptide chains despite their cis-amide propensity, leading to lower entropy costs for the triple-helix formation.

(2) Kinetically, we systematically characterized the kinetic effects of a series of peptoid residues with different cis-amide propensities on the triple-helix folding rate and found that the cis-demanding peptoid residues can drastically slow down the folding. We, therefore, designed a collagen mimetic peptide whose strong triple-helix folding can be controlled on demand by the cis-trans isomerization of its residues. To the best of our knowledge, this is the first of its kind among the various reported CMPs whose folding can be controlled by designed factors (e.g., charges, steric).

(3) Collagen is the primary extracellular component of fibrosis and is extremely challenging to image in disease diagnosis. For applications, we achieved *in vivo* targeting and 3D whole-heart fluorescence scanning of fibrotic remodeling in myocardial infarction with a collagen peptidomimetic probe for the first time.

In conclusion, we hope this manuscript could be considered for publication in *Nature Communications* and provide helpful information for its diverse readership, ranging from protein design, peptidomimetic chemistry, and self-assembly to biomaterials, molecular imaging, matrix biology, and cardiology.

2. It is not clear how cis or trans conformation of amide bond was measured. It seems that it is determined by Ac-X-OMe 1H NMR, however, when the residue is incorporated into collagen sequence, there is evidence to show that the peptoid residue would preserve conformation.

Response and revision:

We thank the reviewer for this important comment.

(1) "It is not clear how cis or trans conformation of amide bond was measured. It seems that it is determined by Ac-X-OMe ¹H NMR..."

Correct, we determined the cis-trans confirmation of the amide bond of peptoid residue X in its monomer state by ¹H NMR of the model compound Ac-X-OMe. As shown in the table below, in the recent three decades, Ac-X-OMe has been widely employed in dozens of studies for evaluating the role of the X-residue's cis-trans propensity in peptides, especially CMPs. Therefore, we used this model to characterize the peptoid residues in our work to be able to correlate and compare with the data in the literature.

X: proline derivatives (Ac-X-OMe)	Structure	$K_{\text{trans/cis}}$	$K_{\text{cis/trans}}$	Reference
Pro		4.6	0.22	J Am Chem Soc. 2001;123(4):777-778. Int J Pept Protein Res. 1994;44(3):262-269.
Hyp		6.1	0.16	J Am Chem Soc. 2001;123(4):777-778. Int J Pept Protein Res. 1994;44(3):262-269.
hyp		2.5	0.40	J Am Chem Soc. 2001;123(4):777-778.
Flp		6.7	0.15	J Am Chem Soc. 2001;123(4):777-778. J Am Chem Soc. 2002;124(11):2497-2505.
flp		2.5	0.40	J Am Chem Soc. 2001;123(4):777-778. J Am Chem Soc. 2002;124(11):2497-2505.
Clp		5.4	0.19	Biopolymers. 2008;89(5):443-454.
clp		2.2	0.45	Biopolymers. 2008;89(5):443-454.
Mep		7.4	0.14	J Am Chem Soc. 2006;128(25):8112-8113.
mep		3.7	0.27	J Am Chem Soc. 2006;128(25):8112-8113.
Mop		6.7	0.15	J Am Chem Soc. 2008;130(10):2952-2953.
(4R) Amp		7.1	0.14	Angew Chem Int Ed Engl. 2014;53(39):10340-10344.
Mcp		4.7	0.21	Angew Chem Int Ed Engl. 2008;47(11):2143-2146.
MCP		5.4	0.19	Angew Chem Int Ed Engl. 2008;47(11):2143-2146.
(4R) Azp		6.1	0.16	J Am Chem Soc. 2010;132(40):13957-13959.
(4R) Acp		5.8	0.17	Angew Chem Int Ed Engl. 2011;50(30):6835-6838.

(4R) Pvp		4.9	0.20	J Am Chem Soc. 2012;134(41):17117-17124.
(4R) Aop		6.7	0.15	J Am Chem Soc. 2017;139(36):12815-12820.
(4R) Oxp		6.7	0.15	J Am Chem Soc. 2017;139(36):12815-12820.
azPro		3.3	0.30	Angew Chem Int Ed Engl. 2019;58(10):3143-3146.
Aze		1.2	0.83	Org Lett. 2020;22(2):348-351.
Pip		5.6	0.18	Org Lett. 2020;22(2):348-351.

Also, similar Ac-X-R small molecule models have been widely utilized to assess the cis-trans amide propensity of peptoid monomers in many peptoid studies.

Model compound (peptoid residue)	Reference
	J Am Chem Soc. 2007;129(29):8928-8929. J Am Chem Soc. 2009;131(45):16555-16567.
	J Am Chem Soc. 2011;133(39):15559-15567. J Am Chem Soc. 2017;139(38):13533-13540.
	J Am Chem Soc. 2007;129(29):8928-8929. J Am Chem Soc. 2009;131(45):16555-16567. J Am Chem Soc. 2012;134(23):9553-9556. J Am Chem Soc. 2019;141(8):3430-3434. Angew Chem Int Ed Engl. 2018;57(33):10549-10553.
	J Am Chem Soc. 2019;141(49):19436-19447.

To clarify what was exactly measured in this study, we made the following changes (in brown) to the first paragraph of the Results section (page 3):

“To investigate the effect of cis-amide propensity on the stability of collagen triple-helix, we introduced a series of peptoid residues at the central X position of a CMP host-guest peptide with the sequence of Ac-(GlyProHyp)₃-Gly-X-Hyp-(GlyProHyp)₃-NH₂ (designated as X-CMP, Fig. 2a,b).²⁴ We measured the circular dichroism (CD) spectra of the X-CMPs (Supplementary Fig. 1a,b) and assessed the X-CMPs’ triple-helical stability (measured by their melting temperature T_m values) via thermal unfolding experiments monitored by CD at 225 nm under a heating rate of 0.5 °C min⁻¹ (See Supplementary Methods, Fig. 2c, Supplementary Fig. 1c,d).²⁵ For each N-gly residue X To measure the inherent cis-trans propensity of each N-gly residue X in its monomer state, we synthesized each model compound Ac-X-OMe and calculated the ratio between its integrated cis- and trans-related peaks in the ¹H NMR spectrum as its $K_{cis/trans}$ value (See Supplementary Methods, Fig. 2a,d, and Supplementary Information Section 3). The Ac-X-OMe

model has been extensively utilized for decades to study the cis-trans propensity of modified prolines and peptoid residues to correlate with the conformation of the sequences featuring these residues, including CMPs.^{12,21,23,26-30} As shown in Fig. 2c and 2d, we found that the Ac-X-OMe compounds' $K_{\text{cis/trans}}$ values of for the peptoid residues we previously reported (*i.e.*, Sar, Nchx, Nleu, Nphe, Nasn, Nlys) ranged from 0.40 to 0.80 and were all higher than Pro ($K_{\text{cis/trans}} = 0.26$), whereas the T_m values of their X-CMPs were almost all greater than that of Pro-CMP, with Nchx-CMP being the most stable (T_m : 64 °C). These data implied that peptoid residues with cis-amide propensity greater than proline can also stabilize the helix."

(2) "...however, when the residue is incorporated into collagen sequence, there is (no) evidence to show that the peptoid residue would preserve conformation...."

We fully agree with the reviewer that the Ac-X-OMe model compound cannot accurately reflect the true cis-amide propensity of the X residues within the peptide chain. In fact, our data strongly supported this point, and this is now one of the main take-home messages of our study (as we have added to the discussion, see below).

Our data demonstrated that this small molecule may not fully model the level of the cis-amide propensity of the X residues in peptides or proteins (Fig. 2&3) and that other supramolecular models may be helpful. For this reason, we adopted X-PP5, a host-guest peptide model for the polyproline helix II (PPII) conformation, to further verify the X residues' amide propensity (Fig. 4). The X-PP5 model was reported by the Neal J. Zondlo group to investigate the PPII propensity of amino acids and the effects of the cis-trans isomerization of the aromatic amino acids (cited as ref 36, 37):

- Brown, A. M.; Zondlo, N. J. A propensity scale for type II polyproline helices (PPII): aromatic amino acids in proline-rich sequences strongly disfavor PPII due to proline–aromatic interactions. *Biochemistry* 2012, 51 (25), 5041-5051.

- Pandey, A. K.; Thomas, K. M.; Forbes, C. R.; Zondlo, N. J. Tunable control of polyproline helix (PPII) structure via aromatic electronic effects: an electronic switch of polyproline helix. *Biochemistry* 2014, 53 (32), 5307-5314.

In most of the previous studies listed in the tables above (pages 8-9 of this letter), the $K_{\text{cis/trans}}$ ratio measured from the Ac-X-OMe compounds was used as the only method to assess the cis-trans propensity of the modified proline residues in peptide folding. In contrast, by combining our Ac-X-OMe and X-PP5 results, we found that while the ratio between the cis- and trans-isomers of Ac-X-OMe can generally reflect the amide preference of the X residue, its actual selectivity in the peptide-bond configuration in proteins may not be as strong as they appear in small molecules, especially in terms of thermodynamics (Fig. 2-4). Therefore, as also suggested by reviewer 3, models that better mimic the structural contexts of peptides and proteins, even multiple ones, are beneficial, but it will require rigorous future studies from the collagen mimetics field to define these models. To emphasize this point, we have made the following comments in agreement with the reviewer in the latest Discussion (page 9).

"We also found that while the ratio between the cis- and trans-isomers of Ac-X-OMe can generally reflect the cis-amide propensity of the X residue, its actual selectivity in the peptide-bond configuration in proteins may not be as strong as it appears in monomers, especially in terms of thermodynamics (Fig. 2-4). As previously reported, although the trans-amide was found to be favored over the cis by 2.5 kcal mol⁻¹ for the small-molecule peptide-bond analog

N-methylacetamide,^{5,61} the difference for an imide bond was calculated to be as low as 0.5 kcal mol⁻¹ within Pro-containing peptides.⁵⁹ This discrepancy may be due to the involvement of multiple structural factors in peptides and proteins (e.g., backbone folding, sidechains) other than the peptide-bond configuration. Because the Ac-X-OMe type of small-molecule models have been widely employed for evaluating the role of the X-residues' cis-trans propensity in peptide and peptoid folding,^{23,30} including CMPs,^{12,13,21,54} our findings caution that these monomer models may not fully reflect the steric and electronic contributions of neighboring residues. Therefore, other assessments, such as those using a trimer (e.g., Gly-X-Hyp) or an oligomer model (e.g., X-PP5), may be desired to objectively determine the overall folding propensity of an X-unit in action, especially for the peptoid residues. Future systematical investigations are required to define the more representative models.”

3. The full spectrum of CD spectrum for those peptoid-containing collagen at a certain temperature (e.g., 25 °C) should be presented. Readers would be able to directly compare the intensity of the CD signatures, from which the stability and conformation of collagen could be predicted. Then it is easier for readers to follow the temperature-dependent stability based on CD in the article.

Response and revision:

We thank the reviewer for this helpful suggestion. We have presented the CD spectra of the X-CMPs in groups in the following Supplementary figure.

(a) The full CD spectra of X-CMPs featuring amino acids, weakly trans-biased N-glys, and strongly cis-demanding N-glys as the guest X unit, recorded in PBS solution (except for N2pic-CMP, which

was measured in 1 mM HCl, pH 3.0). (b) The CD spectra of representative X-CMPs from 215-235 nm. (c) The $K_{\text{cis/trans}}$ values of the X residues and the $[\theta]_{\text{max}}$ (unit: deg cm² dmol⁻¹) as well as λ_{max} values (unit: nm) of the corresponding X-CMP CD curves.

Please also note:

- Because there are too many curves (Supplementary Fig. 1a), we listed all the $[\theta]_{\text{max}}$ and λ_{max} values of the corresponding X-CMP peptides in Supplementary Fig. 1b for the readers to check and compare. (All individual CD spectra were presented in the Supplementary Information, section 4.)
- The reviewer mentioned that the stability and conformation of collagen could be predicted from the CD signatures. Although it is generally true that the CD signal near 225 nm can reflect the stability of the triple-helix, we found that some X-CMPs featuring cis-demanding peptoid residues (e.g., N2pic-CMP) may take up to a month to fully trimerized at 4 °C, despite their high triple-helix stability. In this situation, the CD measurements prior to the complete trimer assembly may yield a lower $[\theta]_{\text{max}}$ value that does not fully correlate with the CMP's high T_m value. Therefore, (also considering the space limit), we put the CD spectra into Supplementary Information (Supplementary Fig. 1).

4. Many groups are working on collagen mimetics, such as Jeffrey D. Hartgerink at Rice University, Michael Yu at Utah University, and David M. Chenoweth at U Penn. Please cite their references in the introduction.

Response and revision:

We thank the reviewer for this excellent suggestion and have added the following references from the groups of Jeffrey Hartgerink, Michael Yu, and David Chenoweth to the introduction of the revised manuscript. Page 2, 3rd paragraph:

“For decades, synthetic collagen mimetic peptides (CMPs), typically featuring the (GPO)_n or (POG)_n (n = 6~10) sequences,¹⁰ have been widely used as models to elucidate the structural features that contribute to collagen stability,¹¹⁻¹⁸ including the effects of the Pro cis-trans isomerization.^{13,19,20”}

New references:

14. Walker, D.R., *et al.* Predicting the stability of homotrimeric and heterotrimeric collagen helices. *Nat. Chem.* **13**, 260-269 (2021).
15. Hulgán, S.A.H. & Hartgerink, J.D. Recent advances in collagen mimetic peptide structure and design. *Biomacromolecules* **23**, 1475-1489 (2022).
16. Zhang, Y., Malamakal, R.M. & Chenoweth, D.M. Aza-glycine induces collagen hyperstability. *J. Am. Chem. Soc.* **137**, 12422-12425 (2015).
17. Li, X., Zhang, Q., Yu, S.M. & Li, Y. The chemistry and biology of collagen hybridization. *J. Am. Chem. Soc.* **145**, 10901-10916 (2023).

Reviewer #3 (Remarks to the Author):

In this article, the authors found that substitution of cis-inducing peptoid monomers in collagen mimetic peptides (CMPs) do not affect the stability of assembled triple helix structures, but they can significantly retard the folding rate. In the case of several cationic cis-inducing monomers, it was found that under acidic conditions (where the monomer is strongly cis-inducing) the folding rate was very slow, but could be significantly accelerated when base was added to deprotonate the side chain (where the monomer is no longer cis-inducing). This property allowed the authors to trigger the assembly of these analogs in the presence of damaged collagen in vivo to image fibrotic lesions caused by myocardial infarction. **The work is carefully performed, and the topic is interesting to the general audience of Nature Communications**, however the key points of this work are not clearly presented.

We thank the reviewer for the favorable comment and the insightful questions.

1. The kinetics and thermodynamics of cis-trans isomerization in peptoids is a well-studied property of peptoids, and prior experimental and computational work in this area is not adequately referenced and discussed. There is also history, dating back to Murray Goodman and others, on the substitution of peptoid residues and related analogs into collagen, and that work is also not adequately referenced or discussed.

Response and revision:

We sincerely thank the reviewer for the helpful comments. We apologize for missing the citation of prior peptoid studies and the works by Murray Goodman. We actually made a statement to dedicate this work to Prof. Murray Goodman in the Acknowledgments of the previous submission (which has been removed in this revised submission due to the journal's formatting requirements).

Edits (in brown) and new citations regarding the previous peptoid studies and the collagen peptidomimetic works by Goodman have now been added to the discussions accordingly. The first paragraph of Discussion (page 9):

“Previous studies on the effect of the peptide bond cis-trans isomerization on protein structures (including collagen) had to rely on proline and modified prolines.^{11-13,18,54} **Because they only span a narrow range of $K_{\text{cis/trans}}$ ratios (e.g., 0.14-0.45),¹³ fresh insights into the structural effects of the cis-preferring amides on collagen (and probably other proteins as well) have been limited. However, as early as 1996, Goodman and co-workers showed that replacing Pro with some peptoid residues such as Nleu within CMPs results in triple-helix stability,⁵⁵ which they attributed to the hydrophobic interchain interactions between the N-gly's sidechain and the adjacent Pro.^{55,56} As a result, perhaps, the effect of the peptoid residue's cis-trans propensity on collagen triple-helicity has never been investigated. Meanwhile, the thermodynamics³¹ and kinetics⁵⁷ of the cis-trans isomerization have been characterized both experimentally and computationally for a variety of monomers in peptoids, many of which exhibit a robust cis-amide propensity.^{23,58} Using these unnatural peptoid residues, our study interrogated, ~~for the first time~~, the residues with a real cis-amide propensity ($K_{\text{cis/trans}} > 1$) and expanded the range of this ratio by an order of magnitude from the proline derivatives [0.26 (Pro) $< K_{\text{cis/trans}} < 10.85$ (Nⁱpr₂ae)] in collagen. We aim to bridge the gap between proline and peptoid residues in protein folding through investigating the amide cis-trans isomerization in**

collagen.”

New references:

55. Goodman, M., Melacini, G. & Feng, Y. Collagen-like triple helices incorporating peptoid residues. *J. Am. Chem. Soc.* **118**, 10928-10929 (1996).
56. Goodman, M., Bhumralkar, M., Jefferson, E.A., Kwak, J. & Locardi, E. Collagen mimetics. *Peptide Science* **47**, 127-142 (1998).
31. Wijaya, A.W., *et al.* Cooperative intramolecular hydrogen bonding strongly enforces cis-peptoid folding. *J. Am. Chem. Soc.* **141**, 19436-19447 (2019).
57. Sui, Q., Borchardt, D. & Rabenstein, D.L. Kinetics and equilibria of cis/trans isomerization of backbone amide bonds in peptoids. *J. Am. Chem. Soc.* **129**, 12042-12048 (2007).
23. Gorske, B.C., Stringer, J.R., Bastian, B.L., Fowler, S.A. & Blackwell, H.E. New strategies for the design of folded peptoids revealed by a survey of noncovalent interactions in model systems. *J. Am. Chem. Soc.* **131**, 16555-16567 (2009).
58. Edison, J.R., *et al.* Conformations of peptoids in nanosheets result from the interplay of backbone energetics and intermolecular interactions. *Proc. Natl. Acad. Sci. U. S. A.* **115**, 5647-5651 (2018).

2. Much of the paper is devoted to the study of the cis/trans propensity of model peptoid monomers (the Ac-X-OMe model), and how this is not correlated to the melting temperature of the triple helix. This is a simple comparison to make, however, the monomer model system is not particularly appropriate since it does not reflect the steric and electronic contributions of neighboring residues, making this correlation – which is the basis for most of the paper – less impactful. A trimer model, like Gly-X-Hyp, would be more representative.

Response and revision:

We fully agree that the Ac-X-OMe model may not be particularly appropriate since it does not reflect neighboring residues' steric and electronic contributions, and a more representative model is needed. In fact, our data supported this point. As we mentioned in the discussion, this is now one of the take-home messages of our study.

However, as shown in the tables we provided in our responses to the 2nd comment from reviewer 2 (pages 8-9 of this letter), in the recent three decades, Ac-X-OMe has been widely employed in dozens of studies for evaluating the role of the X-residue's cis-trans propensity in peptides, especially CMPs. Therefore, we had to employ this monomer model to characterize the peptoid residues in our work to be able to correlate and compare with the data in the literature.

As pointed out by the reviewer, our data well demonstrate that this small molecule may not fully model the level of the cis-amide propensity of the X residues in peptides or proteins (Fig. 2&3), and we think that other supramolecular models may be helpful. For this reason, we have adopted X-PP5, a host-guest peptide model for the polyproline helix II (PPII) conformation, to further verify the X residues' amide propensity (Fig. 4). Indeed, this X-PP5 model was reported to investigate the PPII propensity of amino acids and the effects of the cis-trans isomerization of the aromatic amino acids:

- Brown, A. M.; Zondlo, N. J. A propensity scale for type II polyproline helices (PPII): aromatic amino acids in proline-rich sequences strongly disfavor PPII due to proline–aromatic interactions. *Biochemistry* 2012, *51* (25), 5041-5051.

- Pandey, A. K.; Thomas, K. M.; Forbes, C. R.; Zondlo, N. J. Tunable control of polyproline helix (PPII) structure via aromatic electronic effects: an electronic switch of polyproline helix. *Biochemistry* 2014, 53 (32), 5307-5314.

Combining our Ac-X-Ome and X-PP5 results, we found that while the ratio between the cis- and trans-isomers of Ac-X-Ome can generally reflect the cis-amide preference of the X residue, its actual selectivity in the peptide-bond configuration in proteins may not be as strong as they appear in small molecules, especially in terms of thermodynamics (Fig. 2-4). Therefore, the model remains to be optimized to better mimic the structural contexts of peptides and proteins. We have made this statement in agreement with the reviewer in our latest discussion.

Moreover, we have synthesized a couple of Ac-Gly-X-Hyp-NH₂ trimers and attempted to calculate their $K_{cis-trans}$ ratios of the X-amides.

As can be seen, the ¹H NMR spectrum of the Ac-Pro-Ome monomer is simple, allowing straightforward integration of the cis- and trans-related peaks. (This may partially explain the popularity of this monomer model in the collagen mimetics field.) However, interpreting the overlapping peaks in the ¹H NMR spectrum of the Ac-Gly-Pro-Hyp-NH₂ trimer can be a complex task, possibly requiring two-dimensional NMR spectra first to determine the location of each hydrogen atom. Although we acknowledge the importance of this issue and are also eager to improve the model, we believe that solving the NMR configurations of the Ac-Gly-X-Hyp-NH₂ trimers for over 20 X-residues and verifying whether this trimer model is indeed more representative, are truly a set of topics outside

the scope of this manuscript. We hope the reviewer and editor can understand. Nonetheless, the reviewer indeed pointed out the direction for future investigations. Therefore, we have made the following important edits to the 3rd paragraph in the Discussion (pages 10-11) regarding the need to improve the monomer model.

“...Because the Ac-X-OMe type of small-molecule models have been widely employed for evaluating the role of the X-residues’ cis-trans propensity in peptide and peptoid folding,^{23,30} including CMPs,^{12,13,21,54} our findings caution that these monomer models may not fully reflect the steric and electronic contributions of neighboring residues. Therefore, other assessments, such as those using a trimer (e.g., Gly-X-Hyp) or an oligomer model (e.g., X-PP5), may be desired to objectively determine the overall folding propensity of an X-unit in action, especially for the peptoid residues. Future systematical investigations are required to define the more representative models.”

3. The fact that cis-induction does not destabilize the triple helix, but does impact the rate of folding is fascinating and could be more clearly rationalized and described in kinetic and thermodynamic terms earlier in the paper, with perhaps less emphasis on the monomer model system. For example, does the K_{cis/trans} or the rate of cis-trans isomerization of the peptoid monomer correlate directly with the triple helix folding rate? In other words, does folding only proceed when the peptoid is in the cis conformation? Is amide isomerization rate limiting, or is the cis CMP assembling into the triple helix at a slower rate without amide isomerization)?

Response and revision:

We appreciate the reviewer’s great comments and suggestions. Below are our point-to-point responses and revisions.

(1) *The fact that cis-induction does not destabilize the triple helix, but does impact the rate of folding is fascinating and could be more clearly rationalized and described in kinetic and thermodynamic terms.*

We are glad that the reviewer finds our findings intriguing. We fully agree that this folding behavior should be described in thermodynamic and kinetic terms. We believe that we have already described and rationalized the triple-helical folding of the X-CMPs in thermodynamic terms sufficiently with the DSC data for all types of X residues (Fig. 4, see Results section, starting from page 4, PPII preorganization of the peptoid residues).

However, as the reviewer pointed out, the kinetic analysis was limited in our original submission. Therefore, we have re-analyzed our kinetic data and introduced the folding rate constants accordingly.

Based on a widely cited kinetic study of CMP triple-helical folding by Barbara Brodsky (J Biol Chem. 1999;274(12):7668-7673.), we adopted a third-order kinetic model for all the X-CMPs in our study where the folding of X-CMPs was measured at low concentration (< 0.18 mM):

$$1/[A_t]^2 = 1/[A_0]^2 + 6k_3t$$

where $[A_t]$ is the concentration of the CMP monomer at time t (s) derived from the CD refolding data, and $[A_0]$ is the total peptide concentration. We plotted the refolding curves with $1/[A_t]^2$ versus t for all the X-CMPs we tested (see below, also the new Supplementary Fig. 5a-c).

According to the equation above, we obtained the value of the third-order rate constant (k_3) from the slope of the fitted line for each X-CMP (see table below, also now as the new Supplementary Fig. 5d).

Residue X	$K_{cis/trans}$	$t_{1/2}$ (min)	k_3 ($M^{-2} s^{-1}$)	Residue X	$K_{cis/trans}$	$t_{1/2}$ (min)	k_3 ($M^{-2} s^{-1}$)	Residue X	$K_{cis/trans}$	$t_{1/2}$ (min)	k_3 ($M^{-2} s^{-1}$)
Pro	0.26	21.8	20,000	Pro	0.26	21.8	20,000	Pro	0.26	21.8	20,000
Gly	< 0.01	28.5	11,666	Nchx	0.44	24.9	18,333	N2pic	3.78	>>120	1,500
Ala	< 0.01	22.5	16,666	Sar	0.40	29.9	11,666	Nme ₂ ae	5.32	>>120	833
Leu	< 0.01	22.4	18,333	Nleu	0.43	29.9	11,666	Net ₂ ae	5.49	>>120	833
Phe	< 0.01	46.1	6,666	Nphe	0.64	22.6	16,666	Nbtm ⁺	5.76	>>120	1,333
Asn	< 0.01	36.7	8,333	Nasn	0.73	60.4	5,000	Nme ₃ ae	6.10	>>120	1,000
Lys	< 0.01	35.8	10,000	Nlys	0.80	59.2	5,000	N'pr ₂ ae	10.85	>>120	833

Now, with the kinetic constants added, we can describe the trend and changes in refolding kinetics more precisely. The data clearly showed that whereas the weakly trans-favoring peptoid residues (e.g., from Nchx to Nlys) offer a refolding rate similar to the amino acids, incorporating one strongly cis-favoring peptoid residue can delay the triple-helix folding by over an order of magnitude compared to Pro ($k_3 = 833 M^{-2} s^{-1}$ for Nme₂ae, Net₂ae, and N'pr₂ae, $k_3 = 1,500 M^{-2} s^{-1}$ for N2pic; $k_3 = 20,000 M^{-2} s^{-1}$).

We have added these new data, figures, and tables to the manuscript (see our new Fig. 5e-h and Supplementary Fig. 5) and modified the related text in Results (page 6, 3rd paragraph).

“Next, we conducted the CD refolding kinetic studies of all the X-CMPs at 4 °C after heat-dissociating their triple-helices at 85 °C (see Supplementary Methods) and found similar trends. At the peptide concentration in our tests, the triple-helix folding of the X-CMPs follows a third-order kinetic model,⁴⁷ allowing us to calculate the rate constant (k_3) for each X-CMP (see Supplementary Methods and Supplementary Fig. 5). For the amino acid guest residues, the CD signals of the X-CMPs increased rapidly towards considerable refolding at 120 min (Fig. 5e); for the weakly trans-biased X-peptoid residues, the rise of the CD signal was slightly slower but generally comparable to their amino acid counterparts (Fig. 5e,f). The k_3 rate constant values were also generally comparable between the two groups (Fig. 5e,f). Similar to the results in Fig. 5c, the X-CMPs containing the strongly cis-biased peptoid residues displayed almost horizontal CD curves with minimal rate constants in the protonated state (Fig. 5g) but exhibited noticeable

4-8 times faster refolding after deprotonation of the N-gly sidechains (Fig. 5h). However, such pH- or protonation-responsive changes in refolding rate were not seen for Lys-CMP and other N-glycs whose cis-trans isomerization is not affected by sidechain protonation (e.g., Nleu-, Nchx-, Nphe-CMPs, Supplementary Fig. 6). Altogether, these data demonstrated that the triple-helix folding rates of the trans-favoring X-N-glycs can be comparable to the amino acids, whereas the strongly cis-inducing ones (with $K_{cis/trans}$ values substantially above 1) can drastically slow down the refolding by over an order of magnitude compared to Pro.”

(2) For example, does the $K_{cis/trans}$ or the rate of cis-trans isomerization of the peptoid monomer correlate directly with the triple helix folding rate? In other words, does folding only proceed when the peptoid is in the trans conformation? Is amide isomerization rate limiting, or is the cis CMP assembling into the triple helix at a slower rate without amide isomerization)?

These are great, insightful questions from the reviewer.

Our data strongly indicate that, kinetically, the rate of cis-trans isomerization of the peptoid monomer

directs the X-CMP triple-helix folding as a rate-limiting step. [It's almost impossible to make a direct, quantitative link between the rate of triple-helix folding and the rate of cis-trans isomerization of the peptoid monomer. This is because (1) the amide isomerization is a first-order reaction (rate constant unit: s^{-1}), whereas the triple-helix folding follows a third-order kinetics (rate constant unit: $M^{-2} s^{-1}$) and (2) there is a lack of data on the rate of cis-trans isomerization for all the peptoid monomers.] Nonetheless, the halted refolding for the cis-favoring peptoid residues (Fig. 5c & 5g) in comparison to the normal refolding for the weakly trans-favoring ones (Fig. 5b & 5f), as well as the opposite folding behaviors of the same X-CMPs with different cis-trans amide preferences of just single residue (Fig. 5d), all suggested that the triple-helix folding may only proceed when the peptoid amide bond is in the trans configuration. In other words, the X-CMP is not slowly assembling into the triple-helix with a cis-peptoid amide bond. Overall, the $K_{cis/trans}$ values of the monomers correlate with the triple-helix folding rate in the sense that the folding is promoted when $K_{cis/trans} < 1$ but clearly inhibited when $K_{cis/trans}$ is substantially above 1. However, the exact triple-helix folding rates (k_3) are not quantitatively correlated with the $K_{cis/trans}$ values for individual peptoid residues (see table above on page 17 of this letter), suggesting that further investigations with improved monomer models are needed in the future.

Accordingly, we have added the following sentences to the Discussion (page 11, 2nd paragraph).

“The cis-trans isomerization of proline and its analogs is often key to switching protein conformations and controlling ligand binding.^{1-3,65} In this study, we have proved the concept of controlling the collagen triple-helix folding by switching the cis-trans isomerization of a peptoid residue (Fig. 6). The halted refolding for the cis-favoring peptoid residues (Fig. 5c,g) in comparison to the normal refolding for the weakly trans-favoring ones (Fig. 5b,f), as well as the opposite folding behaviors of the same X-CMPs with the difference in amide preference of a single residue (Fig. 5d), all suggested that the triple-helix folding may only proceed when the peptoid amide bond is in the trans configuration. Overall, the cis-trans isomerization of the peptoid monomer can direct the X-CMP triple-helix folding as a rate-limiting step in the sense that the folding is promoted when $K_{cis/trans} < 1$ but clearly inhibited when $K_{cis/trans}$ is substantially above 1 (Supplementary Fig. 5). Apparently, the cis-demanding peptoid residues can vastly raise the kinetic activation energy barrier of the peptide assembly, but this energy barrier is lowered when the peptoid residues are deprotonated and transformed into trans-amide favoring (Fig. 5d,h). We discovered that the cis-demanding peptoid residues can significantly reduce the folding rate of the triple helix (Fig. 5), suggesting that they can vastly raise the kinetic activation energy barrier of the folding; yet after the peptoid residues are deprotonated and transformed into trans-amide favoring, this energy barrier is lower and the folding is drastically accelerated (Fig. 5d,h). This kinetic property made it possible for the first time to turn the triple-helix formation of a CMP on and off simply by switching the cis-trans propensity of its peptoid residues (Fig. 6d)...”

(3) *The fact ... could be more clearly rationalized and described in kinetic and thermodynamic terms earlier in the paper, with perhaps less emphasis on the monomer model system.*

The reviewer recommends that we focus less on the monomer model system and start describing the triple-helix folding in thermodynamic and kinetic terms earlier in the paper. While we appreciate that this arrangement will allow the readers to reach the conclusive findings sooner, we feel obligated to maintain our original presentation here.

This is mainly because the $K_{cis/trans}$ monomer methodology has been extensively used for decades to

study how the cis-trans propensity of modified prolines and peptoid residues correlate with the conformation of the sequences featuring these residues, including many CMPs and peptoid oligomers (as we have responded to the previous comment and the 2nd comment from reviewer 2). Therefore, the fact that, thermodynamically, the cis-trans propensity of a peptoid residue does not have a dominant effect on the stability of a collagen triple-helix could be more or less a surprise for the researchers who are accustomed to this methodology in our field (in fact, including ourselves in the very beginning). As a result, we think if we don't make this point clear by laying out enough evidence in a stepwise process, our conclusions may not be easily received by a wide audience.

Even so, we surely know the benefit of presenting the conclusive findings sooner and have done our best to keep the first part of the Results as brief as possible (~500 words and two figures). We hope the reviewer and editor can understand.

4. The delicate energetic interplay of local conformation preferences and the thermodynamic stability of folded/assembled structures is **an excellent contribution of this work** and should be put into context with other similar findings in the peptoid field [e.g. Whitelam, PNAS 115, 5647 (2018)].

Response and revision:

We thank the reviewer for the acknowledgment of the contribution of our work and the suggestion. We have incorporated this reference in the discussion. Discussion, 2nd paragraph, page 9:

“... (iii) Every N-gly residue we tested in X-PP5 offered a signature PPII CD signal notably higher than Pro (Fig. 4e), including Nⁱpr₂ae with a $K_{cis/trans}$ value as great as 10.85. These data support that within a Pro-rich sequence, the cis-favoring peptoid residues can not only accommodate the trans-amide configuration but also promote the PPII folding robustly. **Such delicate energetic interplay of local conformation preferences and the thermodynamic stability of the assembled structures has been well demonstrated in the studies of peptoid nanosheets.**^{58,60} Altogether, our data corroborate that...”

New references:

58. Edison, J.R., *et al.* Conformations of peptoids in nanosheets result from the interplay of backbone energetics and intermolecular interactions. *Proc. Natl. Acad. Sci. U. S. A.* **115**, 5647-5651 (2018).

60. Robertson, E.J., *et al.* Design, synthesis, assembly, and engineering of peptoid nanosheets. *Acc. Chem. Res.* **49**, 379-389 (2016).

Reviewers' Comments:

Reviewer #1:

Remarks to the Author:

1. My specific concern regarding the temperature schedule used for thermal denaturation experiments in CD and DSC has been thoroughly addressed and clearly described in the revision. Its good to see that the interpretation was not affected and while the authors note that many other studies incorporate faster melting schemes in their studies, this is an oversight in the other stuides, and in the future if thermodyanmic interpretations are being put forth, reasonable efforts to ensure equilibriuim measurements should be reported. Thank you to the authors for repeating these experiments.

2. My second concern regarding the incorporation of n-pi star interactions in the MD simulation are also properly addressed and again its good to see that the overall interpretation is not affected.

The authors have thoroughly addressed my scientific concerns and I have no additional revisions to request.

Reviewer #2:

Remarks to the Author:

Authors have addressed the comments from reviewers very well.

Reviewer #3:

Remarks to the Author:

I very much thank the authors for adequately addressing all of my concerns. The paper is acceptable for publication. One minor note: please check the number of significant figures used to report the newly measured rate constants. Some of them have 5 significant figures reported (e.g. k3 in table S5), which is too many.

Reviewer #1 (Remarks to the Author):

1. My specific concern regarding the temperature schedule used for thermal denaturation experiments in CD and DSC has been thoroughly addressed and clearly described in the revision. It's good to see that the interpretation was not affected and while the authors note that many other studies incorporate faster melting schemes in their studies, this is an oversight in the other studies, and in the future if thermodynamic interpretations are being put forth, reasonable efforts to ensure equilibrium measurements should be reported. Thank you to the authors for repeating these experiments.

2. My second concern regarding the incorporation of n-pi star interactions in the MD simulation are also properly addressed and again it's good to see that the overall interpretation is not affected. The authors have thoroughly addressed my scientific concerns and I have no additional revisions to request.

Response: We thank the reviewer for accepting our responses and revisions.

Reviewer #2 (Remarks to the Author):

Authors have addressed the comments from reviewers very well.

Response: We thank the reviewer for accepting our responses and revisions.

Reviewer #3 (Remarks to the Author):

I very much thank the authors for adequately addressing all of my concerns. The paper is acceptable for publication. One minor note: please check the number of significant figures used to report the newly measured rate constants. Some of them have 5 significant figures reported (e.g. k_3 in table S5), which is too many.

Response and revision:

We thank the reviewer for this helpful suggestion. We have changed the unit of the third-order rate constants (k_3) from $M^{-2} s^{-1}$ to $\times 10^3 M^{-2} s^{-1}$ and reduced the significant figures of k_3 to no more than 3 as shown in Fig. 5h and Supplementary Fig. 5d (table S5) below.

Fig. 5h:

Supplementary Fig. 5d (table S5):

d

Residue X	$K_{cis/trans}$	$t_{1/2}$ (min)	k_3 ($\times 10^3 M^{-2} s^{-1}$)	Residue X	$K_{cis/trans}$	$t_{1/2}$ (min)	k_3 ($\times 10^3 M^{-2} s^{-1}$)	Residue X	$K_{cis/trans}$	$t_{1/2}$ (min)	k_3 ($\times 10^3 M^{-2} s^{-1}$)
Pro	0.26	21.8	20.0	Pro	0.26	21.8	20.0	Pro	0.26	21.8	20.0
Gly	< 0.01	28.5	11.7	Nchx	0.44	24.9	18.3	N2pic	3.78	>>120	1.5
Ala	< 0.01	22.5	16.7	Sar	0.40	29.9	11.7	Nme ₂ ae	5.32	>>120	0.8
Leu	< 0.01	22.4	18.3	Nleu	0.43	29.9	11.7	Net ₂ ae	5.49	>>120	0.8
Phe	< 0.01	46.1	6.7	Nphe	0.64	22.6	16.7	Nbtm ⁺	5.76	>>120	1.3
Asn	< 0.01	36.7	8.3	Nasn	0.73	60.4	5.0	Nme ₃ ae	6.10	>>120	1.0
Lys	< 0.01	35.8	10.0	Nlys	0.80	59.2	5.0	N ⁱ pr ₂ ae	10.85	>>120	0.8